# Time-resolved multimodal analysis of Src Homology 2 (SH2) domain binding in signaling by receptor tyrosine kinases

Joshua A Jadwin[1†], Dongmyung Oh[2†], Timothy G Curran[3,4], Mari Ogiue-Ikeda[1], Lin Jia[1], Forest M White[3,4], Kazuya Machida[1], Ji Yu[2], Bruce J Mayer[1,2]*

[1]Raymond and Beverly Sackler Laboratory of Molecular Medicine, Department of Genetics and Genome Sciences, University of Connecticut School of Medicine, Farmington, United States; [2]Richard D. Berlin Center for Cell Analysis and Modeling, University of Connecticut School of Medicine, Farmington, United States; [3]Department of Biological Engineering, Massachusetts Institute of Technology, Cambridge, United States; [4]Koch Institute for Integrative Cancer Research, Massachusetts Institute of Technology, Cambridge, United States

*For correspondence: bmayer@uchc.edu

[†]These authors contributed equally to this work

Competing interests: The authors declare that no competing interests exist.

**Abstract** While the affinities and specificities of SH2 domain-phosphotyrosine interactions have been well characterized, spatio-temporal changes in phosphosite availability in response to signals, and their impact on recruitment of SH2-containing proteins in vivo, are not well understood. To address this issue, we used three complementary experimental approaches to monitor phosphorylation and SH2 binding in human A431 cells stimulated with epidermal growth factor (EGF): 1) phospho-specific mass spectrometry; 2) far-Western blotting; and 3) live cell single-molecule imaging of SH2 membrane recruitment. Far-Western and MS analyses identified both well-established and previously undocumented EGF-dependent tyrosine phosphorylation and binding events, as well as dynamic changes in binding patterns over time. In comparing SH2 binding site phosphorylation with SH2 domain membrane recruitment in living cells, we found in vivo binding to be much slower. Delayed SH2 domain recruitment correlated with clustering of SH2 domain binding sites on the membrane, consistent with membrane retention via SH2 rebinding.

## Introduction

Receptor tyrosine kinases (RTK) and tyrosine kinase-associated receptors play an essential role in transducing extracellular signals into the cell. These proteins function as central signaling nodes for a diverse set of normal biological processes including proliferation, differentiation, immune cell activation, neuronal development, angiogenesis, and cell migration. Dysregulated tyrosine kinase activity is also a primary driver of human cancer. Upon activation, RTKs phosphorylate tyrosine residues on themselves and associated proteins, which then serve as binding sites for Src homology 2 (SH2) and phosphotyrosine binding (PTB) domains found within downstream signaling proteins (*Lemmon and Schlessinger, 2010*; *Pawson, 2004*; *Wagner et al., 2013*).

120 SH2 domains have been identified in 110 different proteins (*Liu et al., 2006*; *Tinti et al., 2013*). The central role these proteins play in cellular signaling has made them popular targets for study. The affinities and specificities of many phosphotyrosine (pY)-SH2 interactions have been quantified, and a complex web of downstream pathways initiated by SH2 domain binding has been unraveled (*Hause et al., 2012*; *Huang et al., 2008*; *Jones et al., 2006*; *Liu et al., 2010*; *Tinti et al., 2013*). By contrast, much less is known about the role that spatial and temporal changes in protein

**eLife digest** Individual cells in a multicellular organism must receive signals from the environment and from other cells, and adjust their behavior accordingly. Such signals may cause a cell to grow and multiply, move, or even die. Often these signals are received by receptor proteins, which span the cell membrane and thus provide a way for signals from outside the cell to cause changes inside the cell.

The tyrosine kinases are one such group of membrane receptors. When a signal binds to a tyrosine kinase, the receptor is activated and it can add chemical tags called phosphates to the part of itself, or a neighboring protein, that is inside the cell. These phosphates provide binding sites for other types of proteins, many of which contain a section called a SH2 domain. This transmits the signal and leads to further changes in the cell. However, there are over a hundred different SH2 domain-containing proteins in human cells and we do not have a clear picture of what exactly happens when receptor tyrosine kinases are activated.

Jadwin, Oh et al. have now looked at how the number of SH2 domain binding sites changes over time after a signal is received. The experiments used three different experimental approaches to study a tyrosine kinase called the Epidermal Growth Factor (EGF) receptor, which is often over-active in human cancers. Jadwin, Oh et al. found that the timing of the changes in the number of SH2 domain binding sites on EGF varied widely. The different methods provided different perspectives on exactly when the changes happen, for example, directly observing the binding of SH2 domains to the membrane of living cells under the microscope showed that binding was much slower than expected from other methods that used purified proteins in solutions. This might be due to the receptors taking a relatively long time to form clusters at the membrane after they receive a signal.

Further experiments suggested that what happens when EGF is activated may depend not only on the number of SH2 domain binding sites made, but also the timing and the physical arrangement of those sites. A long-term goal for further studies is to understand how various types of signals can lead to different outcomes in the cell.

phosphorylation play in signal transduction. Current experimental approaches, however, have the potential to address system dynamics directly.

Methods such as quantitative mass spectrometry (MS) and live cell fluorescence microscopy are now capable of tracking temporal changes in cellular physiology across remarkably short time steps. For instance, isotopic protein-tagging MS methods, including SILAC and iTRAQ, allow us to quantify the relative and absolute abundance of pY sites found within hundreds of proteins across multiple time points (*Curran et al., 2015*; *Olsen et al., 2006*). Data from these experiments has been used to map temporal changes in molecular signaling, providing us with a more comprehensive understanding of pathway dynamics (*Olsen et al., 2006*; *Zheng et al., 2013*). At the same time, high resolution microscopy, coupled with more traditional imaging methods and biochemical studies, has allowed us to begin to dissect the spatial reorganization of RTKs and their effectors following receptor activation (*Abulrob et al., 2010*; *Chung et al., 2010*; *Endres et al., 2013*; *Grusch et al., 2014*; *Hsieh et al., 2010*; *Morimatsu et al., 2007*; *Needham et al., 2013*; *Sako et al., 2000*).

We previously used single particle tracking photoactivated localization microscopy (sptPALM) (*Manley et al., 2008*) to visualize and quantify individual SH2 domain binding events at the plasma membrane in response to receptor activation (*Oh et al., 2012*). From these studies we developed a model in which the high density of tyrosine-phosphorylated sites on the membrane results in the repeated rebinding of SH2 domain-containing proteins before they can escape into the cytosol. This local rebinding suppresses the apparent off-rate and prolongs the membrane dwell time of SH2 domain-containing proteins. Furthermore, we showed that clustering of SH2 binding sites, a well-known consequence of RTK activation (*Abulrob et al., 2010*; *Endres et al., 2013*; *Ichinose et al., 2004*; *Sergeev et al., 2012*), further suppressed the apparent off-rate of the GRB2 SH2 domain, suggesting that signal output could be influenced by changes in physical properties such as phosphite distribution.

Traditional two-dimensional signaling diagrams and phosphopeptide-based affinity studies fail to fully capture the complexities and importance of spatial and temporal changes in receptor activation, protein phosphorylation, and SH2 domain-containing protein recruitment. Indeed, existing biochemical and proteomic methods each have strengths and technical limitations, such that no single approach provides a clear and unambiguous perspective. Thus to better understand RTK signaling dynamics, we have employed three orthogonal experimental techniques: SH2 domain-based reverse-phase binding assay (far-Western blotting) to assess global changes in SH2 binding sites, iTRAQ (isobaric tagging for relative and absolute quantification)-based phosphotyrosine-specific MS to identify changes in the abundance of specific phosphopeptides, and live cell single particle tracking using total internal reflection fluorescence (TIRF) microscopy to assay recruitment of SH2 domains to the membrane in vivo (see *Figure 1*). Combining these complementary approaches to analyze the EGF response in a single, well-characterized experimental system revealed previously uncharacterized properties of the EGFR signaling network. We quantified and compared binding site creation kinetics for a large set of SH2 domains with high temporal resolution, and were able to define a previously unappreciated role for phosphosite clustering in sculpting the dynamic profile of signal output downstream of tyrosine kinase activation.

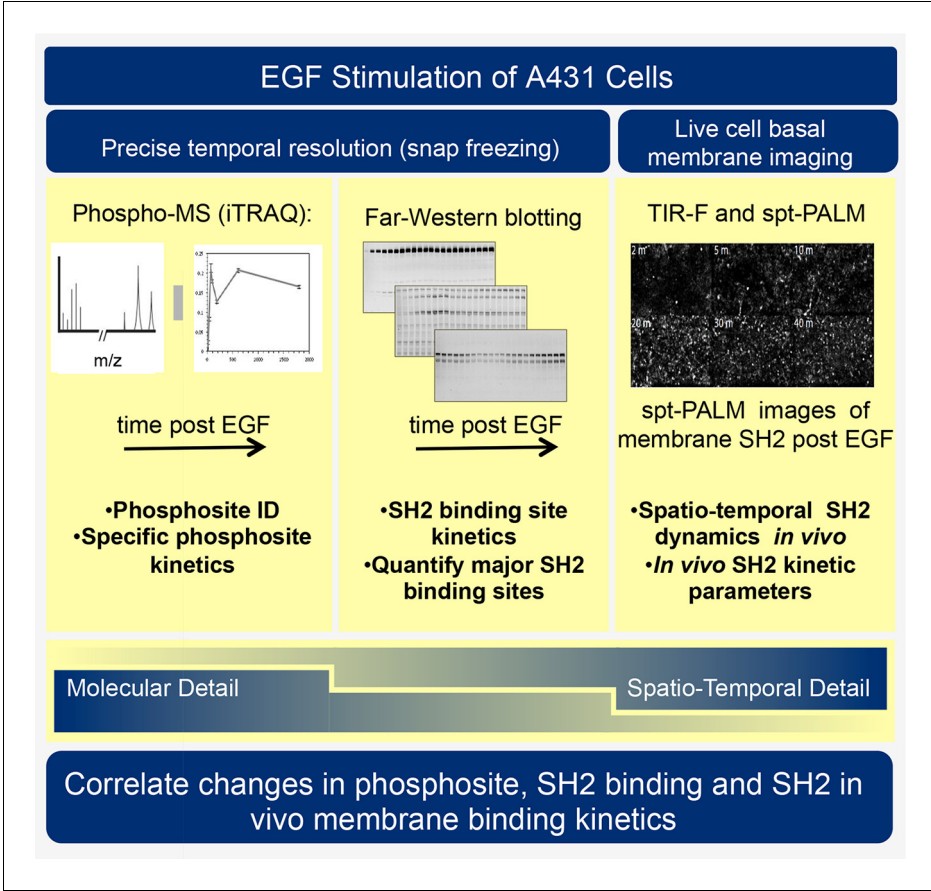

**Figure 1.** Experimental outline. Human A431 cells were stimulated with EGF and dynamic responses analyzed using three complementary methods. Samples snap-frozen at different time-points after EGF treatment were analyzed by mass spectrometry (MS) to detect relative changes over time in abundance of specific tyrosine-phosphorylated peptides. Parallel samples were analyzed by far-western blotting with a panel of SH2 domain probes to visualize changes over time in binding sites for each SH2 domain. Finally, living cells expressing fluorescent SH2 domains were imaged to quantify membrane binding dynamics for each SH2 domain.

## Results

### Dynamic EGF-dependent changes in tyrosine phosphorylation

We first performed far-Western blotting, a reverse-phase SH2 binding assay (*Machida et al., 2007*) to quantify changes in the binding sites for multiple SH2 domains across the set of phosphoproteins in EGF-stimulated A431 cells, an EGFR-overexpressing squamous-cell carcinoma cell line (*Stanton et al., 1994*; *Wrann and Fox, 1979*). Cells were starved overnight, stimulated with 25 ng/ml EGF, flash frozen at multiple time points post-stimulation, and lysates run on LDS-PAGE in duplicate. All blots included positive and negative pY controls to assess non-specific binding. Blots were probed with recombinant GST-tagged SH2 and PTB probes and an anti-pY antibody (*Supplementary file 1*) (*Machida et al., 2007*). Of the probes tested, 27 were selected for further analysis based on significant, reproducible, and dynamic binding to EGF-stimulated A431 cell lysates (*Figure 2A*, *Figure 2—figure supplement 1*). Anti-pY blots of the same lysates showed a rapid increase in total phosphorylation upon EGF treatment, dominated by a major band corresponding to EGFR (*Figure 2A*, top panel). Blots also contained a number of minor bands whose phosphorylation varied over time. SH2 binding patterns varied across the set of SH2 domains tested, however the molecular weights of major bands were consistent, suggesting that most SH2 probes bound predominantly to a relatively small set of highly phosphorylated proteins (*Figure 2A*, *Figure 2—figure supplement 1*). Most prominent were five highly dynamic bands, which were identified by immunodepletion (*Figure 2—figure supplement 2A–D*) and exhibited three well-defined kinetic patterns. These major bands were (1) EGFR, a ~195 kDa band whose phosphorylation increased rapidly following EGF treatment, dipped slightly and then remained relatively constant; (2 and 3) the focal adhesion protein p130CAS, a doublet at 150 and 115 kDa whose phosphorylation was high in unstimulated cells, decreased rapidly upon EGF treatment, and then rebounded at later time points; (4) the scaffolding protein GAB1, a single band at 130 kDa that was phosphorylated with kinetics similar to EGFR, then rapidly returned to near basal levels; and (5) the scaffold/adaptor SHCA, a relatively weak band at 71 kDa which displayed kinetics similar to that of EGFR (*Figure 2A*, *Figure 2—figure supplement 1*, *Supplementary file 2*).

### Quantification of SH2 domain binding sites

Far-Western provides a means to characterize the global binding patterns of multiple SH2 domains across multiple cellular states (*Machida et al., 2010*; *2007*). To compare quantitative changes in binding patterns of different SH2 domains upon EGF treatment, we performed unsupervised hierarchical clustering on SH2 domain binding data over the 60-min EGF stimulation time-course (*Figure 2B*). Six distinct SH2 domain-binding clusters ($R^2$>0.85) were identified, which were broadly consistent with literature-reported interactions. For example, GRB2 SH2 and SHCA PTB domains most strongly interacted with EGFR (*Blagoev et al., 2003*; *Okutani et al., 1994*), SHP2 and p85α/β SH2 domains interacted most strongly with GAB1 (*Holgado-Madruga et al., 1996*), and focal adhesion-associated proteins such as CRK, CRKL, NCK1 and NCK2 bound most strongly to p130CAS (*Rivera et al., 2006*; *Sakai et al., 1994*). All but three SH2 domains (RASGAP C-terminal, NCK1 and NCK2) showed at least some binding to EGFR. As expected, probes from homologous proteins clustered together (i.e. p85 α/β). However, SH2 probes isolated from proteins with two SH2 domains showed more variation (e.g. RASGAP-N, -C and -NC) (*Supplementary file 2*).

Tyrosine phosphorylation in EGF-stimulated A431 cells is dominated by phospho-EGFR (see *Figure 2A*, top panel), and as a result most SH2 domains displayed strong EGFR binding. To better understand the relative affinity of each SH2 for various phosphoproteins, we normalized the quantified SH2 binding signal to pY abundance (from anti-pY immunoblot) and reclustered the data (*Figure 2—figure supplement 3A*). After normalization, EGFR no longer dominated binding. Only three domains (GRB2, SHCA PTB and to a lesser extent ARG) displayed a greater relative affinity for EGFR than would be expected from motif-independent pY binding (i.e. anti-pY blotting). Instead, most SH2 domains displayed higher relative affinity for GAB1 and p130CAS. This analysis is designed to highlight SH2 binding specificity for particular protein targets, whereas the non-normalized FW results reflect a combination of binding specificity and the abundance of binding sites, and thus are more likely to predict the level of binding to different proteins in vivo.

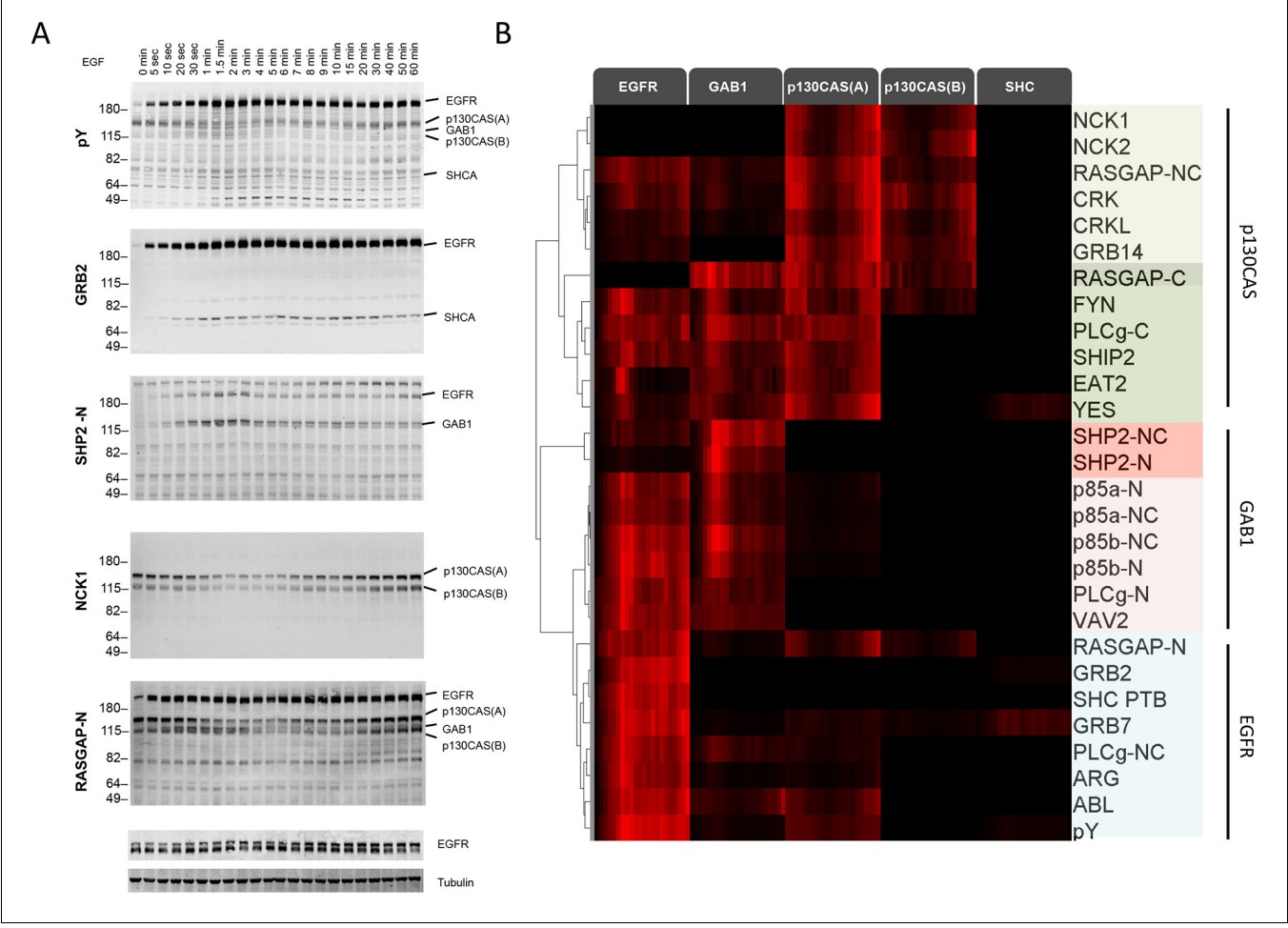

**Figure 2.** Dynamic EGF-dependent changes in tyrosine phosphorylation patterns revealed by SH2 domain far-Western analysis. (**A**) Representative anti-pY Western (upper panel) and far-Western blots (next four panels) of 60-min EGF stimulation time-course. Far-Westerns using GRB2, SHP2-N, NCK1 and RASGAP-N are shown to illustrate major binding patterns identified (see **B**). Additional SH2 blot data are provided in *Figure 2—figure supplement 1*. Immunoblotting with antibodies to EGFR and tubulin was used to confirm equal loading. (**B**) Hierarchical clustering of SH2 domains on the basis of binding to four major phosphoproteins (EGFR, GAB1, p130CAS, and SHCA). Signal was normalized to maximum band intensity across all time points and all bands for each probe replicate. Then data for each phosphoprotein was averaged in a probe specific manner (red represents greater percentage of total signal, max = 1, min = 0). Names of SH2/PTB domain probes are indicated on the right. Colored boxes represent SH2 clusters defined by un-centered correlation coefficient >0.85.

The following figure supplements are available for figure 2:

**Figure supplement 1.** Representative far-western blots using 1 PTB and 26 SH2 domain probes for EGF-stimulated A431 cells.

**Figure supplement 2.** Identification of major SH2 domain binding bands by immunodepletion.

**Figure supplement 3.** Relative SH2 binding specificity and pY EGFR-SH2 interaction kinetics.

## pY-EGFR specific SH2 domain binding and downstream signaling

In principle, if different sites on a particular protein were phosphorylated with very different kinetics, different SH2 domains might bind to that protein with different kinetics. For the most part, however, we observed only modest differences in the temporal pattern of the binding of different SH2 domains to particular phosphoproteins. To better visualize the temporal variation in EGFR binding, SH2 domains were clustered based on changes in phospho-EGFR binding over time (*Figure 2—figure supplement 2B*). The majority of probes, including GRB2, ARG, p85 and the SHCA PTB, displayed

dynamics similar to that of total EGFR phosphorylation, with a rapid and sustained increase in binding. However, a set of SH2 domains including CRK, CRKL, RASGAP and GRB14 showed a more gradual increase in phosphorylation over the 60-minute time-course. And a set of four SH2 domains including EAT2, SHP2-N and the only two Src-family kinases (SFKs) tested, YES and FYN, displayed a rapid increase in binding followed by a rapid and sustained decrease (*Supplementary file 2*).

To test whether this variation was due to binding of SH2 domains to specific EGFR sites phosphorylated with different kinetics, we performed EGFR phosphosite-specific Western blotting and compared site-specific dynamics to those obtained by FW. While phosphospecific antibodies may not in all cases be totally specific for the sites against which they were raised, they are widely used to assess changes in particular phosphosites such as those in EGFR. Qualitatively, the phosphorylation dynamics of specific sites mirrored those of specific SH2 domains by far-Western (*Figure 2—figure supplement 3B*). In particular, the kinetics of EGFR pY992, which is located within a canonical CRK binding motif (pYXXP), strongly correlated with CRKL probe EGFR binding kinetics ($R^2$=0.88). Both displayed an initial rapid rise to approximately half maximum, followed by a slow increase to maximum over the 60-minute time-course (*Figure 2—figure supplement 3D*). Strong correlation was also observed between the dynamics of GRB2 SH2 domain binding and the GRB2 binding site EGFR pY1068 ($R^2$=0.91) (*Figure 2—figure supplement 3E*) (*Okutani et al., 1994*). SHCA PTB domain binding correlated well with total EGFR pY ($R^2$=0.89), consistent with a recent report suggesting that the SHCA PTB binding site on EGFR, pY1148, dominates receptor phosphorylation (*Curran et al., 2015*) (*Figure 2—figure supplement 3F*). However, not all binding data could be predicted from known phosphosite specificities. For example, the far-Western binding kinetics of YES, FYN and EAT2 qualitatively resembled the rapid and transient phosphorylation of EGFR pY974 and pY1045, providing evidence for potentially unappreciated interactions (*Figure 2—figure supplement 3B*, *Supplementary file 2*).

Finally in order to understand the relationship between SH2 binding site creation and downstream signaling we assessed the activation time-course of ERK1 and ERK2, which are known to be activated after EGF stimulation via recruitment of proteins including SHP2, GRB2 and SHCA to EGFR (*Roskoski, 2014*). Activating phosphorylation of ERK1/2 reached a maximum approximately 4 min after EGF stimulation, compared with 1.5–2 min for tyrosine phosphorylation of EGFR (*Figure 2—figure supplement 3G*).

## Quantitative phospho-specific mass spectrometry

Although SH2-based far-Western blotting provides a unique insight into overall patterns of binding for different SH2 domains, it is highly dependent on phosphosite abundance, and does not identify specific phosphorylated sites. To obtain a complementary view of EGF-induced changes in tyrosine phosphorylation, iTRAQ MS was used to quantify the phosphorylation kinetics of individual phosphosites. In contrast to FW blotting, MS provides relative abundance data across a set of samples for specific phosphopeptides that can be attributed to a particular site on a particular protein; however it provides little information about the abundance of any particular peptide relative to all other phosphopeptides in the sample. A431 cells were flash frozen and lysed at eight representative time points (0, 10 s 30 s, 1 min, 1.5 min, 3 min, 10 min, 30 min) following EGF treatment (25 ng/ml, as for FW blotting). Prior to MS analysis, anti-pY immunoblotting was performed to ensure that phosphorylation kinetics and band patterns of MS samples were similar to those used for FW analysis (*Figure 3—figure supplement 1*). Lysates were then digested with protease, enriched for tyrosine-phosphorylated peptides by anti-pY immunoprecipitation, and purified using affinity chromatography prior to analysis by reverse phase liquid chromatography tandem mass spectrometry (LC-MS/MS). Relative phosphorylation levels of specific peptides were quantified by iTRAQ analysis using three biological replicates; only those phosphopeptides identified in all eight time points, in at least two replicates at each time point, were included in our analysis. MS experiments were run in data-dependent discovery mode, in order to maximize network coverage in a relatively unbiased fashion.

In total, 132 unique pY sites were identified from 93 proteins (*Supplementary file 2*). Of the phosphosites identified, 88 (67 proteins) displayed an EGF-dependent increase in tyrosine phosphorylation (defined by at least one time point with a two-fold or greater increase in abundance and a statistically significant increase in at least two time points following EGF treatment), including those from EGFR, GAB1 and the SH2-containing proteins SHCA (SHC1), PLCγ1, SHP2 and CRKL (*Supplementary file 2*). The percent of EGF-dependent phosphosites indentified in this study (65%)

was higher than in previous studies using cell lines with more moderate EGFR expression (*Blagoev et al., 2004*; *Olsen et al., 2006*; *Zhang et al., 2005*). Among EGF-dependent sites identified in this study, 64 (57 proteins) were not found within a curated database of EGF-dependent phosphosites (*Hornbeck et al., 2012*) (*Supplementary file 2*).

Gene ontology (GO)-based functional analysis of EGF-responsive phosphoproteins identified enrichment for terms associated with positive regulation of signaling and peptidyl tyrosine modification (*Supek et al., 2011*). EGF-nonresponsive sites, on the other hand, tended to be associated with regulation of cell adhesion and locomotion. However, a large number of terms were shared by the two protein sets, indicating significant functional overlap (*Figure 3A*). By comparing the amino acid frequencies surrounding phosphotyrosines for each group, we found that EGF-nonresponsive sites were more likely to contain the CRK SH2 binding motif (pYXDP/L), consistent with the functional association of these proteins with adhesion and locomotion. EGF-responsive peptides however, displayed little sequence consensus (*Figure 3B*).

## Phosphopeptide analysis of FW-identified proteins

iTRAQ analysis identified multiple sites from the four major phosphoproteins detected by FW (*Figure 3C–F*, *Supplementary file 2*). Comparison of MS and FW data for these proteins revealed both similarities and differences. Four unique EGFR pY peptides and one tyrosine/serine dually phosphorylated peptide were identified (*Figure 3C*). The tyrosine-only sites tended to display a continual increase in abundance over the first 10 min, compared to the more rapid plateau in signal intensity seen in FW and anti-pY (*Figure 2*, *Supplementary file 2*). The dually phosphorylated peptide varied significantly over the time-course and was not scored as EGF-dependent. Five unique phosphopeptides were identified for GAB1, all of which increased rapidly, but did not show the rapid dephosphorylation seen in anti-pY blots and FW (*Figure 3D*). Like the EGFR pY peptides, the single SHCA phosphopeptide, a known GRB2 binding site (*Harmer and DeFranco, 1997*), displayed slower phosphorylation kinetics than those obtained from anti-pY and FW blots (*Figure 4E*). MS phosphopeptide data also did not recapitulate the p130CAS dephosphorylation and rephosphorylation pattern seen on anti-pY immunoblots and SH2 far-Westerns. Instead, the six p130CAS phosphopeptides identified by iTRAQ MS trended toward an increase in phosphorylation following EGF stimulation, though only 2 of the 6 displayed a statistically significant change in abundance (*Figure 3F*, *Supplementary file 2*). Overall, FW blotting and MS-based quantification of individual sites provided rather different perspectives on tyrosine phosphorylation dynamics, highlighting the importance of using complementary experimental approaches. These results suggest that changes in the binding of an SH2 domain to a multiply phosphorylated protein can be difficult to predict precisely from changes in the phosphorylation of individual phosphosites on that protein.

Finally, most phosphopeptides showed a transient decrease in phosphorylation at the 3 min time point, similar to the dip seen on far-Westerns and anti-pY Westerns, especially for EGFR and SHCA, at approximately 4 min (*Supplementary file 2*). The fact that a similar pattern is seen using two very different experimental approaches increases confidence that it accurately reflects the actual behavior of the system.

## In vivo membrane recruitment of SH2 domains

FW and MS data revealed dynamic changes in potential SH2 binding sites in response to EGF, but provide little insight into the actual binding kinetics of SH2 domain-containing proteins in living cells. To address this issue, we used live-cell microscopy to quantify the recruitment of SH2 domains to the basal membrane. A similar set of 25 SH2 domains to that used for FW blotting, each fused with the photoactivatable fluorescent protein tdEOS, were expressed individually in A431 cells. TIRF microscopy (*Video 1*) and sptPALM were used to reveal SH2 localization patterns and to quantify membrane binding rates and diffusion constants (D) following EGF treatment (*Supplementary file 2*, *Table 1*). It is important to note that all SH2-containing proteins contain additional interaction domains that may affect the recruitment kinetics of the full length proteins in vivo; however in order to directly compare results with SH2 binding site data from the FW and MS studies above, we assayed in vivo binding using isolated, ectopically expressed SH2 domains. In the case of GRB2, we previously showed that full-length GRB2 and the SH2 domain alone were very similar in their in vivo

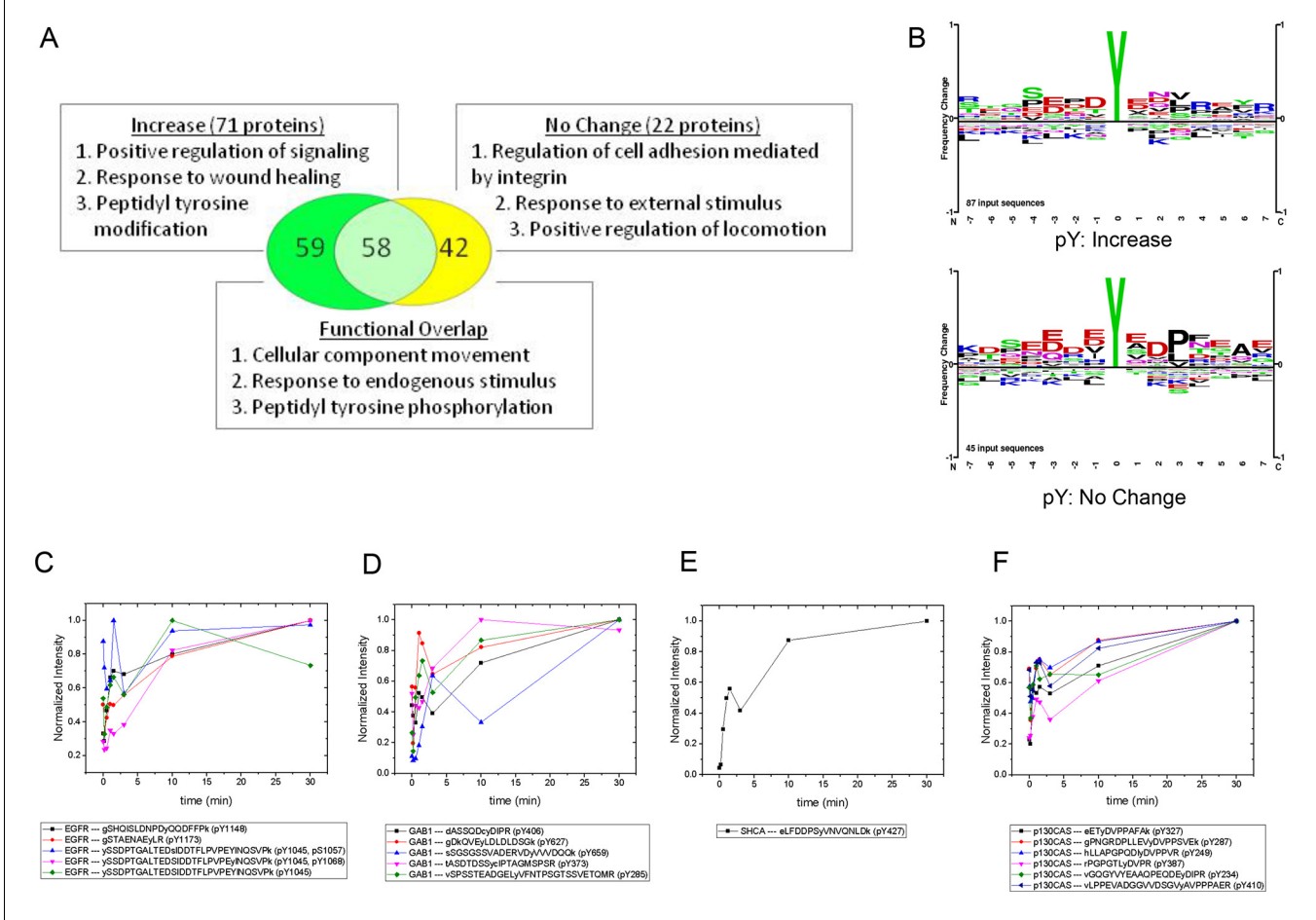

**Figure 3.** MS analysis of tyrosine-phosphorylated peptides in EGF-treated cells. (**A**) Venn diagram showing overlap of significant gene ontologies for proteins containing peptides whose phosphorylation was enhanced or unchanged by EGF (p≤0.05, Bonferroni corrected). The number of unique or overlapping ontologies observed for each protein set is indicated within the diagram. GO terms listed represent the three largest GO parent terms returned by REVIGO (*Supek et al., 2011*). (**B**) Amino acid frequency logos for sites whose phosphorylation was enhanced (upper) and unchanged (lower) by EGF stimulation. Background data is PhosphoSitePlus pY database. (**C–F**) Relative phosphopeptide abundance for peptides derived from EGFR (**C**), GAB1 (**D**), SHCA (**E**) and p130CAS (BCAR1) (**F**). Specific phosphopeptide sequences are listed. Results are average of three biological replicates.

The following figure supplement is available for figure 3:

**Figure supplement 1.** Anti-pY blot of three biological replicates analyzed by iTRAQ phospho-specific MS.

membrane binding behavior (*Oh et al., 2012*), indicating that for this protein the kinetics were dominated by SH2-pY interactions.

Live-cell imaging of EGF-induced SH2 domain membrane binding revealed a wide variety of dynamic behaviors among the different SH2 domains tested. However, a number of trends were apparent in light of the binding specificities identified by FW hierarchical clustering. SH2 domains that predominantly bound GAB1 (e.g. SHP2-N, SHP2-NC) tended to reach maximum binding rapidly, diffuse rapidly upon binding, and display a relatively diffuse spatial distribution before stimulation and a mix of clustered and diffuse localization after stimulation. EGFR-binding SH2 domains (e.g. GRB2 and GRB7) displayed relatively slow recruitment kinetics and diffusion rates, and their binding sites tended to cluster together into discrete foci in response to EGF. p130CAS-binding domains (e.g. NCK, CRK, RASGAP-NC) generally had either slower recruitment kinetics or exhibited small to no net recruitment following stimulation; they also primarily localized to structures

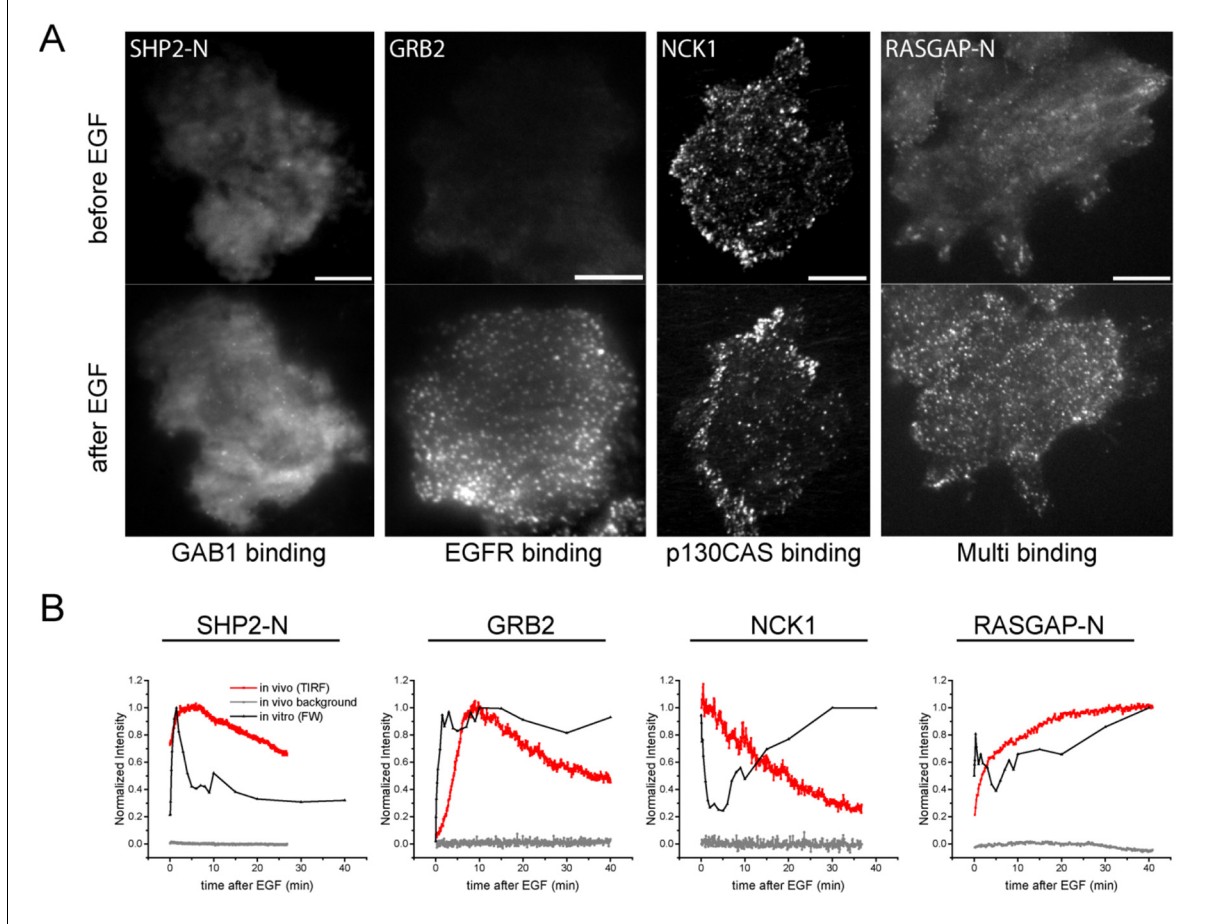

**Figure 4.** In vivo localization and recruitment kinetics of SH2 domains. (**A**) Representative total internal reflection fluorescence (TIRF) microscopy images of fluorescently tagged SH2 domains before and 40 min after EGF stimulation, for SHP2-N, GRB2, NCK1 and RASGAP-N SH2 domains. Scale bars = 10 μm. (**B**) Comparison of change in total membrane SH2 fluorescence from imaging live cells (*red*) and change in FW-based SH2 binding (*black*) following EGF stimulation. *Gray* lines indicate TIRF background signal. Data is normalized to maximum. See ***Supplementary file 2*** for complete dataset. FW data represent average of multiple technical replicates; in vivo data are from single representative experiments.

The following figure supplements are available for figure 4:

**Figure supplement 1.** Analysis of in vivo SH2 domain localization and membrane binding.

**Figure supplement 2.** Quantification of EGF binding.

**Figure supplement 3.** Quantification of GRB2 SH2-tdEOS and pY-EGFR in A431 cells.

**Figure supplement 4.** Linear response of FW assay.

**Figure supplement 5.** EGF-induced cell growth of A431, Cos1, and H226 cells.

**Figure supplement 6.** Quantification of Grb2 binding sites and in vivo Grb2 SH2 recruitment in A431 cells stimulated with 1 ng/ml EGF.

**Figure supplement 7.** NCK SH2 and p130CAS display similar localization patterns.

**Figure supplement 8.** CRK FW time-course blots from A431 and COS1 cells simulated in parallel with 10 ng/mL EGF.

**Table 1.** Quantification of in vivo SH2 binding dynamics and binding site kinetics. To compare in vivo imaging and FW data directly, for each SH2 domain in vivo membrane recruitment kinetics and timecourse of total binding (to all bands) by FW were fit to the first order exponential recovery function $(1-e^{-t/\tau})$, where $\tau$ is the time constant and t is time after EGF. *denotes data that fit poorly to the recovery function (R-square <0.5). n = number of biological replicates used for calculation of $\tau$ from in vivo imaging. SEM of $\tau$ values from multiple replicates shown in parentheses. D is the diffusion constant of SH2 molecules on the membrane in cells stimulated by EGF, measured by single-molecule tracking technique 40 min post-EGF (see Materials and methods). For each SH2, data from >3000 trajectories in a single cell were used to calculate D. ND = not determined.

| SH2 Domain | in vivo (single molecule and TIRF imaging) | | | in vitro (FW) |
| --- | --- | --- | --- | --- |
| | Rcruitment time $\tau$ (min) | n | D ($\mu m^2$/s) | Phosphorylation time $\tau$ (min) |
| SHIP2 | 0.55 (0.15) | 3 | 0.040 | 1.13 (0.81)* |
| SHP2-N | 0.91 (0.02) | 2 | 0.187 | 0.60 (0.01) |
| GRB14 | 0.99 (0.47) | 2 | ND | -1.50 (0.08) |
| SHP2-NC | 1.57 (0.18) | 5 | 0.081 | 1.57 (0.18) |
| SHP2-C | 1.85 (0.49) | 8 | 0.759 | ND |
| EAT2 | 2.26 (1.10) | 2 | 0.097 | 0.64 (0.51)* |
| PLCγ1-NC | 2.44 (0.18) | 2 | 0.022 | 0.62 (0.19) |
| p85α-NC | 3.06 (0.32) | 2 | 0.004 | 0.51 (0.09) |
| SHC PTB | 3.25 (0.08) | 4 | 0.007 | 0.69 (0.08) |
| GRB7 | 3.45 (0.67) | 3 | 0.009 | 0.77 (0.05) |
| VAV2 | 4.27 (0.52) | 2 | 0.034 | 0.80 (0.17) |
| SHC SH2 | 4.57 (0.20) | 4 | 0.026 | ND |
| GRB2 | 4.58 (0.41) | 11 | 0.021 | 0.59 (0.03) |
| PLCγ1-N | 5.15 (2.28) | 2 | 0.022 | 0.75 (0.33) |
| CRK | 5.54 (0.64) | 2 | 0.016 | -2.80 (0.29) |
| RASGAP-NC | 5.92 (0.80) | 2 | 0.076 | -1.27 (0.54)* |
| PLCγ1-C | 6.48 (1.27) | 5 | 0.043 | 0.30 (0.03) |
| ARG | 6.55 (0.90) | 2 | 0.010 | 0.54 (0.20) |
| p85α-N | 6.88 (0.78) | 2 | 0.016 | 0.67 (0.10) |
| RASGAP-N | 6.99 (2.53) | 2 | 0.011 | -2.51 (0.98) |
| FYN | constant | 2 | 0.008 | 0.43 (0.04) |
| NCK1 | decrease | 2 | 0.009 | -0.83 (0.03) |
| RASGAP-C | constant | 2 | ND | 1.27 (0.54)* |
| ABL | constant | 3 | ND | 0.70 (0.40) |
| YES | ND | | ND | -3.62 (1.48)* |

resembling focal adhesions (*Figure 4A*, *Figure 4—figure supplement 1A–C*, *Supplementary file 2*, *Table 1*).

For comparison, we also analyzed the kinetics of binding of fluorescently labeled EGF under the same conditions. We used oblique illumination excitation in order to minimize the background signal from soluble EGF (*Teramura et al., 2006*), while still allowing us to assess binding to the whole cell (instead of just the basal surface as in TIRF illumination). The kinetics of EGF binding were very similar to those seen for anti-pTyr and FW blots of EGF-treated cells, peaking at ~1.5 min (*Figure 4—figure supplement 2*; compare with *Supplementary file 2*)

To compare SH2 recruitment kinetics of different SH2 domains, time constants were calculated for the recruitment curves obtained for each SH2 domain. Membrane fluorescence measurements following EGF treatment were plotted with respect to time and fitted to the first order exponential recovery function, $1-e^{(-t/\tau)}$, where t denotes time after EGF stimulation. The time constant ($\tau$) is proportional to the time required to reach half-maximal binding. SH2 domain membrane recruitment time constants negatively correlated with their effective mobility (diffusion rates) (*Table 1*, *Figure 4—*

**Video 1.** Real time imaging of GRB2 SH2-tdEOS membrane recruitment under TIR microscopy using 488 nm excitation. GRB2 SH2-tdEOS recruitment to the basal membrane of A431 cells begins immediately following EGF. The total number of recruited molecules reaches equilibrium at ~10 min, as indicated by the fluorescence intensity profile (right, 1 frame = ~10 s).

figure supplement 1D). In other words, those SH2 domains that were recruited most quickly tended to diffuse more rapidly after binding.

One surprising result was that for most SH2 domains, in vivo binding kinetics lagged significantly when compared to the kinetics of SH2 binding site creation, as measured by far-Western (*Supplementary file 2*, *Figure 4B*, *Table 1*). On average, time constants calculated from membrane fluorescence intensity were ~6 times greater than time constants for the same domains calculated from far-Western blotting, and in vivo time constants had a much greater range when compared with those obtained by FW (*Table 1*). This lag was unexpected, as simple calculations using best-estimate SH2 concentrations and rate constants ($k_{on}$ and $k_{off}$) suggested that binding and unbinding should equilibrate rapidly, within seconds.

Using the GRB2 SH2 as an example, we examined several potential mechanisms that could give rise to this discrepancy, all of which of turned out to be unsubstantiated. First, unexpectedly low concentrations of GRB2 SH2-tdEOS and phosphorylated EGFR in vivo could prolong the time to EGF-induced equilibrium. However, direct biochemical measurements of GRB2 SH2-tdEOS and pY-EGFR in these cells (an average of 6.5 μM and 1.5 μM, respectively) did not support this theory (*Figure 4—figure supplement 3A–D*). We also tested the possibility that SH2 binding site creation on the basal membrane (where in vivo binding is measured by TIRF) might be much slower than on the apical membrane. Our results from quantifying confocal microscopy of EGF-stimulated cells immunostained with anti-pY antibody (*Figure 4—figure supplement 3E–G*) indicated that this is not the case. Finally, we considered the possibility that the FW blotting and anti-pY signals were saturated, thereby obscuring increases in phosphosites after 1.5 min. We addressed this by running serial dilutions of samples on the same blot and probing with Grb2 SH2 domains (*Figure 4—figure supplement 4*). As can be seen, the signal was essentially linear until the highest concentration of lysate (which was twice the amount used for the blots shown in *Figure 2* and quantified in *Supplementary file 2*). Thus if there were any increase in abundance of phosphorylated sites in the lysate after 1.5 min, it would have been easily detected by the assay.

We also considered the possibility that the discrepancy in reaching maximal phosphorylation and maximal in vivo binding could be related to the very high level of EGF receptor expression in A431 cells, and to the high concentration of EGF used for stimulation (25 ng/ml is approximately 4 nM). It has been previously reported that stimulation of A431 cells with this concentration of EGF is in fact growth inhibitory (*Barnes, 1982*; *Gill and Lazar, 1981*; *Lifshitz et al., 1983*). We therefore tested a much lower EGF concentration, 1 ng/ml, which is mitogenic for A431 cells in our hands (*Figure 4—figure supplement 5*). As shown in *Figure 4—figure supplement 6*, when A431 cells were stimulated with 1 ng/ml EGF, the kinetics of EGFR tyrosine phosphorylation assessed by anti-pY and FW blotting were similar to those seen at 25 ng/ml, although the overall EGFR phosphorylation levels were much lower. In cells stimulated by 1 ng/ml EGF, in vivo recruitment of the Grb2 SH2 domain was again slower than expected from the abundance of Grb2 binding sites, though the difference was less, approximately two-fold slower vs. ~7.5-fold at 25 ng/ml.

It is also worth noting that most p130CAS-binding domains showed increased membrane binding in response to EGF, even as total binding site availability (as measured by FW) declined over the first 5–10 min (*Figure 2*, *Supplementary file 2*). This is likely the result of the relatively high concentration of EGFR phosphosites in the EGF-stimulated A431cells. One exception however was NCK1 SH2, which did in fact display an initial decrease in membrane binding upon EGF treatment, though it did not exhibit the late increase (10–60 min) or occur at the same rate seen in FW (*Figure 4B*, *panel 3*, *Supplementary file 2*). Composite images created from sptPALM movies showed both the NCK1 SH2 domain and p130CAS localizing to focal adhesion-like structures on the cell periphery (*Figure 4—figure supplement 7*). Taken together, these results suggest that the SH2 domain binding in vivo is a phosphosite-dependent, but not fully equilibrated, process.

## Changes in total cellular phosphorylation correlate with the apparent SH2 on-rate

If SH2 membrane binding in vivo is indeed far from equilibrium, then maximum phosphorylation and maximum binding need not be coincident. Instead, maximum phosphorylation may be more closely correlated with the maximum *rate* of recruitment, i.e. $\gamma_{on}$ (t) = $k_{on}$(t)[pY][SH2], where $\gamma_{on}$ (t) is the apparent on-rate at time t after addition of EGF. To test this we directly measured the rate of binding by utilizing sptPALM to count the number of new SH2 molecules appearing at the membrane during a small time window (*Das et al., 2015*). For these experiments, we utilized the GRB2 SH2 domain to minimize effects from binding to non-EGFR phosphoproteins (*Figure 2*). We found that the binding rate of GRB2 SH2 increased much more rapidly (τ=2.08 min) than total binding (*Figure 5A*, *black line*). To corroborate this rate measurement, we attempted to recapitulate the experimentally obtained changes in GRB2 SH2 abundance at the membrane by combining apparent on-rate measurements, with apparent off-rate measurements (λ$_{off}$) calculated using a previously published method (*Figure 5A*, *red line*) (*Oh et al., 2012*). This relationship can be represented as follows:

$$\frac{\mathrm{d}[\mathrm{mem:SH2}]}{\mathrm{dt}} = \gamma_{on}(t) - \lambda_{off}(t) \cdot [mem:SH2](t) \tag{1}$$

The solution of this nonhomogeneous first order differential equation being:

$$[\mathrm{mem:\ SH2}](t) = \mathrm{c}e^{-\int \lambda_{off}(t)\mathrm{dt}} + e^{-\int \lambda_{off}(t)\mathrm{dt}} \int \gamma_{on}(t) \cdot e^{\int \lambda_{off}(t)\mathrm{dt}}\mathrm{dt} \tag{2}$$

Plugging in the measured values for $\gamma_{on}(t)$ and $\lambda_{off}(t)$ for GRB2 SH2 returns a SH2 membrane binding [Mem:SH2] curve with a relatively slow rate of recruitment. This rate was similar to that obtained experimentally (*Figure 5B*, compare with *Figure 4B*), indicating that SH2 recruitment is likely kinetically controlled and not an equilibrated process.

## GRB2 SH2 domain binds rapidly to non-clustered sites

We previously reported that clustering of SH2 binding sites was associated with a decrease in the apparent membrane dissociation rate (increase in dwell time) (*Oh et al., 2012*). We proposed that this was due to increased SH2 rebinding to phosphosites that were more closely packed upon clustering. It was therefore plausible that clustering might also play a role in the apparent delay in maximal recruitment of SH2 domains to membrane binding sites. Consistent with this idea, analysis of GRB2 SH2 cluster size and cluster number in EGF-treated cells showed that cluster formation reaches a maximum at 10–15 min (*Figure 5C*), a time-scale coincident with that of maximal recruitment of GRB2 SH2 to the cell membrane. Furthermore, as mentioned above, we found that the recruitment time constants for individual SH2 domains negatively correlated with their diffusion rates, which we previously reported to depend on the extent of phosphosite clustering (*Oh et al., 2012*). These results suggested that maximal membrane recruitment of SH2 domains may lag behind the generation of phosphorylated SH2 binding sites due to the relatively slow clustering of those sites.

To assess more directly the role of clustering in SH2 recruitment kinetics, we counted the number of molecules detected within the cluster regions on sptPALM images. We found that the aggregated binding rate within clusters increased at a relatively a slow pace, reaching maximum at approximately 10.5 min, with a time constant of τ$_{luster}$ = 4.05 min (*Figure 5D*, *red*). In contrast, the binding rate in non-cluster regions, obtained by subtracting cluster-associated binding events from the total binding events detected, reached a maximum at approximately 2.5 min (τ$_{non-cluster}$ = 1.08 min) (*Figure 5D*, blue). These results strongly suggest that binding to non-clustered phosphosites is relatively fast and tracks with the level of phosphosites available, while increases in in vivo binding seen at later points (after 1–2 min) can be attributed to increased binding to clustered sites as they accumulate.

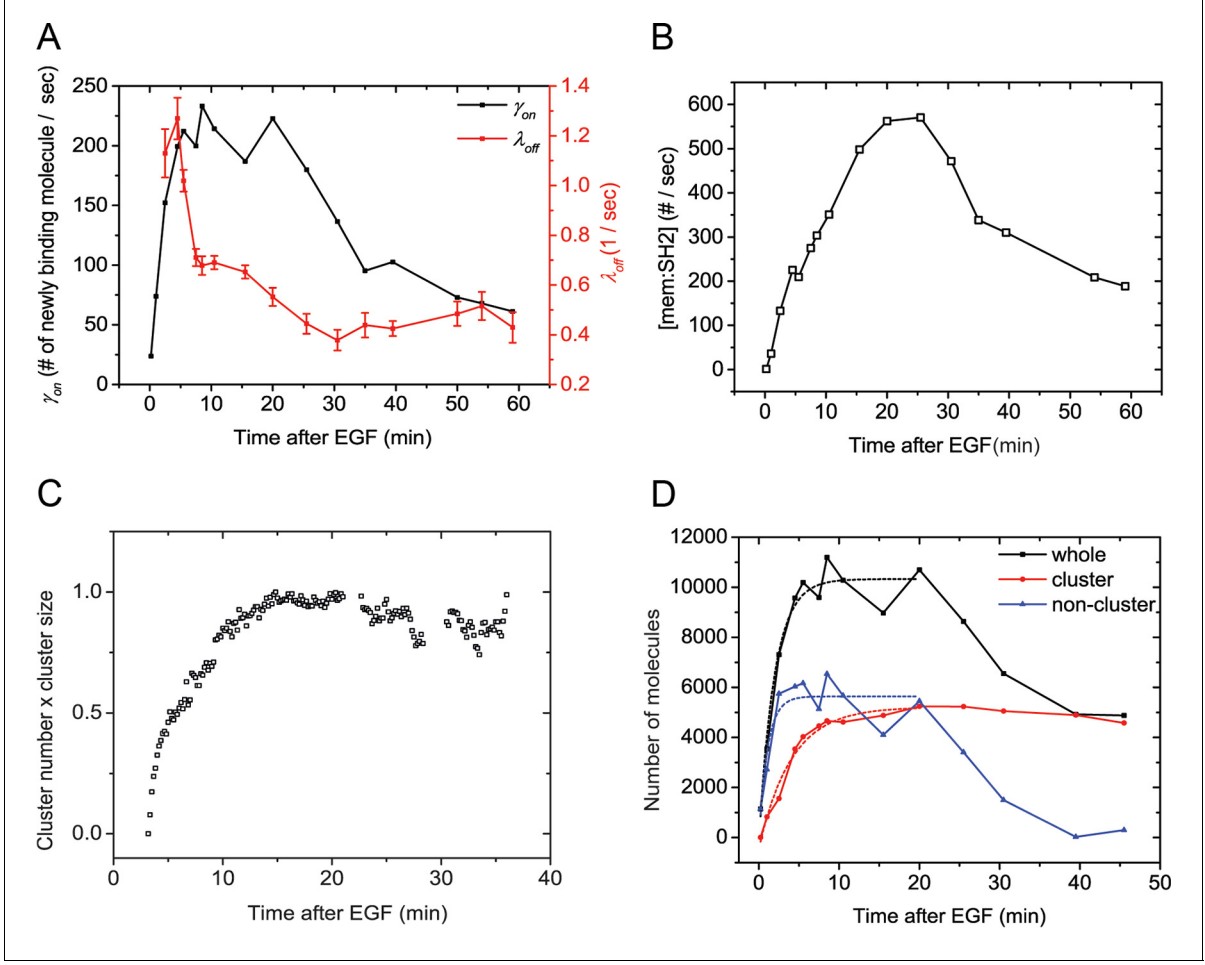

**Figure 5.** Quantification of Grb2 SH2 membrane recruitment rate in EGF-stimulated cells. (A) SptPALM measurement of apparent membrane recruitment rate ($\gamma_{on}$, *black line*) and apparent membrane dissociation rate ($\lambda_{off}$, *red line*) following stimulation with EGF. $\gamma_{on}$ is computed as the total number of observed recruitment events divided by the timespan (0.8 min). $\lambda_{off}$ is computed based on the membrane dwell-times of the observed molecules as previously described (*Oh et al., 2012*). Errorbars on $\lambda_{off}$ represent statistical sampling errors due to the finite numbers of single-molecule trajectories used for this calculation. (B) SH2 membrane binding curve calculated using experimentally determined $\gamma_{on}$ and $\lambda_{off}$ values. (C) Kinetics of GRB2 SH2 binding site clustering (cluster size x cluster number) after EGF treatment. (D) Number of newly recruited GRB2 SH2 molecules (*black, whole*), and those within clusters (*red, cluster*) and outside of clusters (*blue, non-cluster*) after EGF stimulation. Dotted lines show fit with exponential recovery function.

## Pervanadate-treated cells display reduced GRB2 binding site clustering and rapid recruitment

To further validate the relationship between SH2 recruitment and EGFR clustering, we sought a way to measure GRB2 SH2 domain recruitment in the absence of clustering. Binding of EGF to EGFR has been shown to induce dimerization, as well as higher order clustering of the receptor in a kinase-activity-dependent manner (*Abulrob et al., 2010*; *Endres et al., 2013*; *Ichinose et al., 2004*; *Sergeev et al., 2012*). We attempted to bypass EGF-induced receptor multimerization by treating cells with the competitive irreversible tyrosine phosphatase inhibitor pervanadate (PV) (*Huyer et al., 1997*), which increases cellular pY levels by blocking dephosphorylation. PV treatment led to a rapid increase in phosphotyrosine on a large number of proteins as shown by anti-pY immunoblot, but FW blotting with the GRB2 SH2 domain demonstrated that EGFR was by far the major GRB2 SH2 binding protein in PV-treated cells (*Figure 6*). Note that at later time points, PV-treated cells had significantly higher levels both of total pY, and of Grb2 SH2 binding sites on EGFR, than were achieved at any time after EGF treatment. We confirmed the lack of GRB2 SH2 binding site clustering in PV-

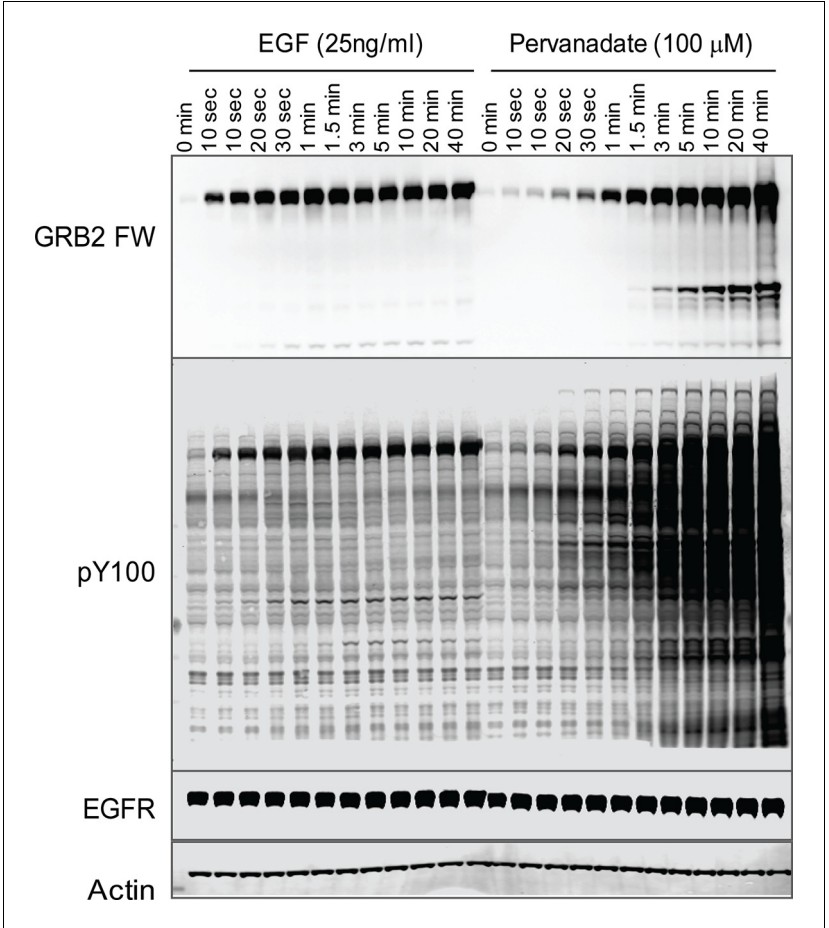

**Figure 6.** Tyrosine phosphorylation in EGF treated vs. pervanadate (PV) treated cells. Representative GRB2 SH2 far-Western and anti-pY (PY100) immunoblots for EGF and PV treated cells. Immunoblotting with antibodies to EGFR and actin was used to confirm equal loading.

treated cells using a variety of measures. Firstly, GRB2 SH2-tdEOS showed a diffuse spatial distribution in PV-treated cells (*Figure 7A*, *Video 2*). Furthermore, individual membrane-associated GRB2 SH2 molecules were much more mobile (i.e. had a higher diffusion rate) in PV-treated cells than in cells treated with EGF (*Figure 7B* and *Video 2*). These results were consistent with a lack of GRB2 SH2 binding site clustering in PV-treated cells.

In the absence of apparent binding site clustering, GRB2 SH2 membrane recruitment in PV-treated cells occurred rapidly ($\tau$=0.81 +/- 0.15 min) (*Figure 7C*). Indeed, in contrast to EGF treated cells, where in vivo binding lags behind the rate of binding site creation as measured by Grb2 FW, in PV-treated cells the rate of membrane binding was comparable to the rate of GRB2 binding site creation (*Figure 7D*). These results are consistent with the interpretation that delayed SH2 domain binding kinetics in EGF-treated cells are the result of increased binding site clustering over time.

## Discussion

In this study we employed three methods to monitor EGF-induced tyrosine phosphorylation and SH2 domain binding in the EGFR overexpressing A431 cell line: far-Western blotting, iTRAQ MS, and in vivo imaging of plasma membrane SH2 domain recruitment. These methods proved to be complementary and far from interchangeable, as response kinetics showed surprising differences between methods. Nevertheless, by combining and comparing data from each technique we were able gain unique insights into the relationship between protein tyrosine phosphorylation, creation of

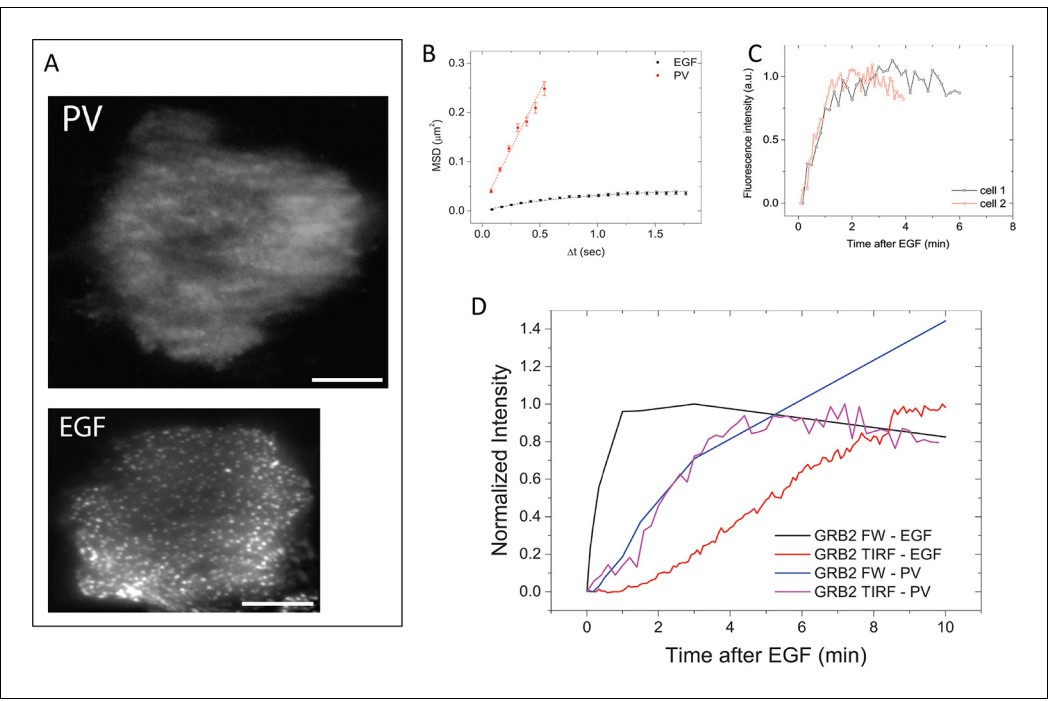

**Figure 7.** GRB2 SH2 recruitment dynamics in pervanadate (PV) treated cells. (**A**) TIRF microscopy images of fluorescently tagged GRB2 SH2 in pervanadate-treated (PV)- and EGF-stimulated cells (40 min post-stimulation). Scale bars = 10 µm. (**B**) Mean square displacement (MSD) of GRB2 SH2 in EGF- (*black*) and PV-treated cells (*red*). Results based on ~8000 (PV) and ~10,000 (EGF) pooled single-molecule trajectories from two cells. Error bars represent SEM. (**C**) Temporal progression of Grb2 SH2 recruitment rates (apparent on-rate) for two PV-treated cells. (**D**) Comparison of GRB2 SH2 recruitment kinetics (TIRF fluorescence imaging, n=2, and GRB2 FW (n=2) total binding, in either EGF-treated (in vivo) or PV-treated cells (in vivo). FW data is normalized so that the maximum signal in EGF treated cells equals 1.

SH2 domain binding sites, and membrane recruitment of SH2 domains in response to EGF, as well as into the capabilities of each method.

FW data provided an unprecedented global overview of changes in binding sites for 27 representative pY binding domains in response to EGF. A unique aspect of the FW approach is that it provides insight into the overall binding preferences of particular SH2 domains among all the tyrosine phosphorylated proteins in a cell lysate. Our results are generally consistent with previously reported SH2 domain-phosphoprotein interactions. On the whole, however, EGFR binding dominated SH2 probe binding patterns in our experiments. For example, probes such as CRK and CRKL, which are generally considered specific for focal adhesion proteins, displayed significant EGFR binding. This is likely due in part to the high EGFR expression level in A431 cells. When the CRK SH2 domain was used to probe far-Western blots of EGF-stimulated COS1 cells, which express more typical amounts of EGFR, strong p130Cas binding was seen in the absence of significant binding to EGFR (*Figure 4—figure supplement 8*). Thus, as expected from simple binding kinetics, both SH2 domain binding specificity and the concentration of SH2 domain binding sites determine the complexes formed and thus the output of

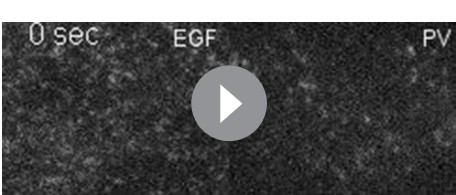

**Video 2.** GRB2 SH2-tdEOS mobility in EGF- and PV-treated cells following EGF stimulation. GRB2 SH2-tdEOS molecules on the basal membrane of A431 cells treated with 25 ng/ml EGF (left) and 100 µM PV (right). Videos were recorded 40 min after addition of EGF or PV at 0.1 Hz. GRB2 SH2-tdEOS can be seen diffusing significantly faster in PV treated cells and its movement along the membrane is not constrained.

tyrosine kinase signaling pathways. This is consistent with the idea that signaling pathways can be 're-wired' by dysregulated RTK activity (*Jones et al., 2006*; *Pawson and Warner, 2007*).

By temporally resolving both short and long stimulation time points, we were also able to show that the timecourse of phosphorylation can differ when specific EGFR sites are compared. Differences observed in SH2 binding site kinetics by far-Western blotting were mirrored by similar differences in EGFR phosphosite-specific immunoblotting. Thus the stoichiometry of SH2 and PTB domain-containing effectors bound to EGFR is temporally regulated by differential phosphorylation/dephosphorylation of their specific binding sites following stimulation with EGF. However, our global SH2 binding analyses indicate that these differences were generally modest, and as such, evaluation of their importance for EGFR signaling will require a more in-depth analysis. Nevertheless, for multiply phosphorylated proteins, such as EGFR, far-Western likely provides a more complete picture of SH2-mediated signaling than phosphosite-specific Western blotting alone, because many SH2 domains display at least some affinity for multiple phosphosites on these proteins, and the relative stoichiometry of phosphorylation of different sites is variable and is challenging to quantify (*Curran et al., 2015*; *Jones et al., 2006*).

Using iTRAQ MS we were able to show that the phosphorylation state of over 65% of observed pY peptides was significantly altered following EGF stimulation, a much higher percentage than detected in previous studies using similar significance thresholds in cell lines with more moderate EGFR expression. Many of these sites are not listed as EGF-dependent within the PhosphoSitePlus database. Again, these data are consistent with the idea that EGFR overexpression is associated with a significant expansion of its classical downstream signal transduction pathways. However, a more systematic analysis is needed to quantify the effects of receptor expression level on EGFR signalosome plasticity. iTRAQ MS was also able to identify functional differences in protein tyrosine phosphorylation. GO analysis revealed significant differences between EGF-dependent and -independent phosphoproteins. As expected, proteins with EGF-dependent phosphosites were associated with ontologies related to activation of pY-dependent growth factor signaling. On the other hand, proteins containing non-affected sites tended to be associated with ontologies related to integrin signaling and cellular adhesion. Phosphosites in these unaffected proteins were also more likely to contain CRK or NCK binding sites, consistent with their role in cell adhesion signaling.

The relatively fine temporal resolution of our analyses allowed us to capture small dynamic changes in protein tyrosine phosphorylation that were likely missed by previous studies using broader time intervals. Of particular note is the damped oscillation in the tyrosine phosphorylation levels soon after stimulation with EGF. The phenomenon was reproducibly captured for EGFR and to a lesser extent GAB1 by anti-pY and far-Western blotting. The observation of a similar dip in most MS-detected phosphosites suggests that it is a general feature of EGF-induced tyrosine phosphorylation and not a methodological artifact. While we can only speculate about the specific mechanism, it is likely due to feedback mechanisms that function rapidly and are directly linked to EGFR kinase activity (e.g. recruitment of SH2-containing phosphatases) (*Ferrell and Ha, 2014*). Moreover, such phosphorylation-dependent phosphatase-mediated negative feedback could also explain why the phosphorylation of SH2 domain-containing proteins, such as SHCA, reached equilibrium long before their recruitment to the membrane reached a maximum. EGF-dependent recruitment or activation of phosphatases might serve to counter the effect of the continued increases in binding between SH2-containing proteins and EGFR, as detected by TIRF imaging.

We also observed a number of inconsistencies between MS and far-Western in phosphorylation kinetics. This was particularly true for the focal adhesion scaffold p130CAS, where individual phosphosites did not show the decline in abundance after EGF treatment seen in FW (and anti-pY) blots. Similarly, we also observed noticeable differences in the rate of phosphorylation of sites on EGFR when compared to matched anti-pY Westerns. How can different analytic methods give such different results for the same samples? It is important to realize that both FW and anti-pY blots integrate the total signal for all phosphosites on a given protein, which can vary widely both in their affinity for different SH2 domains and in their stoichiometry of phosphorylation. Discrepancies can arise when the relatively limited subset of phosphosites detected by MS is not phosphorylated and/or dephosphorylated with the same kinetics or to the same level as the totality of sites that contribute to Western or far-Western signals. This is more likely to be an issue for proteins with many potential phosphorylation sites, such as EGFR and p130CAS. More directed analyses of specific multiply phosphorylated proteins will be needed to resolve this question definitively. Nevertheless, these findings

highlight the advantage of using multiple orthogonal methods to gain a full and accurate picture of the molecular events downstream of RTK activation.

Perhaps the most notable and surprising finding from this study was the difference between the kinetics of SH2 binding site creation, as measured by FW, and in vivo membrane recruitment, as measured by live cell imaging. The average time required for membrane binding to reach a maximum following EGF stimulation was nearly six times longer than expected based on the abundance of binding sites, indicating that SH2 membrane recruitment is not in rapid equilibrium with tyrosine phosphorylation. The extent of the apparent lag in in vivo binding correlated to some extent with SH2 binding specificity and the diffusion rate of bound SH2 domains—in general, the lag was shorter for SH2 domains that bound mostly to GAB1.

A number of factors must be taken into account when considering rates of effector binding in the cell. Of course activation and phosphorylation of the receptor are triggered by binding of ligand (EGF), which has been extensively studied both experimentally (*Waters et al., 1990*; *Wiley, 1988*) and theoretically (*Lauffenburger and Linderman, 1996*) over the years. As discussed below, ligand binding can be diffusion-limited (*Berg and Purcell, 1977*; *Lagerholm and Thompson, 1998*) under conditions where receptor concentration is high, particularly when density is not homogeneous (that is, receptors are clustered). In principle, this can affect both extracellular ligands (EGF) and cytosolic ligands (SH2 domain proteins). When we examined the kinetics of EGF binding, it closely tracked with EGFR phosphorylation by anti-pTyr and FW (*Figure 4—figure supplement 2*), as expected. However, SH2 recruitment lagged behind EGFR phosphorylation, likely due to EGFR clustering.

We recently reported that the effective off-rate of an SH2 domain from EGFR receptor is slower than expected due to diffusion-limited kinetics, as clustering favors phosphosite hopping or rebinding of the SH2 domain at the membrane (*Oh et al., 2012*). Such a slowdown in off-rate would necessarily be accompanied by a similar effect on the on-rate (*Berg and von Hippel, 1985*), the combination of which may be the root cause for the observed slow SH2 recruitment. Indeed, several of our results support such a notion. In particular, the results presented here suggest that clustering plays a role in the apparent delay in reaching maximal SH2 binding in vivo, and that this delay may be a natural feature of the interaction between SH2 domain-containing proteins and the membrane. Consistent with this hypothesis, the time-course of GRB2 SH2 binding site clustering in response to EGF was remarkably similar to the time-course of membrane recruitment (*Figures 4B,5C*). This makes sense, because clustering of pY binding sites should significantly slow down diffusion-limited reaction rates. Moreover, when GRB2 binding sites were created without inducing clustering (by PV treatment), the apparent delay was no longer seen, and the rates of GRB2 binding site formation and membrane recruitment were nearly equal (*Figure 7D*). Furthermore, when we used sptPALM to measure the recruitment of new domains to the membrane directly (equivalent to the effective membrane on-rate), this rate reached a maximum much more quickly than did total membrane binding. In particular, the kinetics of new molecule binding to unclustered sites were very similar to the kinetics of binding site generation measured by far-Western (*Figure 5D*). This strongly suggests that the apparent delay in total membrane binding observed in vivo is the result of changes in the kinetics as a result of clustering, and not an issue of binding site availability alone.

While the qualitative observations support the effects of diffusion-limited kinetics, quantitative modeling of the observed kinetics is not straightforward. A simple back-of-the-envelope estimation of the diffusion-limited on-rates can be obtained by two methods. First, given the ratio of the diffusion-limited off-rate, which we previously measured to be on the order of 1 sec$^{-1}$ and the $K_d$, which is about 1 μM, suggests an effective on-rate of about $10^6$ M$^{-1}$sec$^{-1}$. Another way is to look at the upper limit of the diffusion-limited on-rate, which is about $4\pi Ds/N$, where $D$ is the diffusion constant ($\sim$1 μm$^2$sec$^{-1}$), $s$ is roughly the size of the clusters, and $N$ the number of pYs in the cluster, which would give an on-rate of 0.1–1 x $10^6$ M$^{-1}$sec$^{-1}$. In either case, these estimates both suggest that diffusion-limited binding would result in a delay of seconds and not minutes as we observed in the experiment. Therefore, other unidentified factors may be at play here. For example, these simple analyses ignored the spatial heterogeneity of the pY distribution, and potential competition of binding to many different phosphoproteins as well as competition between different cellular compartments. Fully resolving this question clearly requires more elaborate quantitative models.

This novel observation of a lag in reaching maximal recruitment to phosphorylated sites in vivo raises the question of how such a delay might impact signaling. Signal outputs are subject to multiple positive and negative feedback loops, making it difficult to assess the specific role of such a

phase delay. For example, our results show that ERK activation (assayed by ERK1/2 phosphorylation) reaches a maximum at ~4 min, between the time of maximal GRB2 binding sites (1–2 min) and maximal recruitment of GRB2 to the membrane (~10 min). It is possible that the sustained, clustering-dependent increase in SH2 binding might be involved in regulating the duration of ERK activation. Another interesting possibility is that clustering-dependent recruitment is a mechanism to diversify the response kinetics upon receptor stimulation. For example, recruitment of effectors to GAB1, a relatively freely diffusing scaffold protein, reaches a maximum much more rapidly than recruitment of effectors to EGFR itself. Phosphorylation of scaffolding and adaptor molecules such as GAB1, SHCA, and others may play a general role in increasing the spatiotemporal diversity of receptor output (*Kholodenko et al., 2010*; *Zheng et al., 2013*).

Our results also raise the question of whether the delayed in vivo binding kinetics are a general property of RTK signaling, or if they are specific to situations where the density of activated receptors is extremely high, such as A431 cells stimulated with high concentrations of EGF as in our studies. We did compare the kinetics of Grb2 binding in vivo compared with the generation of Grb2 binding sites by FW for a much lower EGF concentration (1 ng/ml), and found that in vivo binding was delayed, but that the difference between rates of maximal binding sites and maximal in vivo binding was less (roughly two–three fold, vs. ~eightfold at 25 ng/ml EGF). However, we note that at 1 ng/ml EGF, the level of SH2 recruitment to membranes in vivo is near the detection limit of our imaging method. Indeed, most previous studies of RTK signaling have used high levels of ligand in order to maximize detection of output signals such as tyrosine phosphorylation. Thus additional experiments will be required to determine the conditions where clustering will have significant effects on rates of downstream signaling, and to develop a robust theoretical framework to describe this effect.

Taken together, the data presented here provide a strong rationale for a multimodal approach to the analysis of PTM-mediated signaling. Such an approach allows the identification of areas of discrepancy, which can then be explored by more directed experimentation. This is analogous to the interplay between in vivo experimentation and computational modeling, where new insight is driven by discrepancies between model predictions and experimental results (*Kitano, 2002*; *Verveer and Bastiaens, 2008*). In the process of reconciling seemingly discordant results obtained by interrogating the same system with multiple methodologies, one can gain novel insights into the system not possible from studies dependent on a single methodology.

## Materials and methods

### SH2 domain expression constructs

Characteristics of SH2 domain constructs used in this study are provided in *Supplementary file 1*. GST-SH2 probes for far-Western were cloned in pGEX backbones as previously described (*Machida et al., 2007*) and are available through Addgene at https://www.addgene.org/Bruce_Mayer/. SH2-tdEOS clones were generated using Gateway cloning as previously described (*Oh et al., 2012*).

### Antibodies and reagents

Anti-pY immunoblots and immunofluorescence experiments were performed using mouse monoclonal pY100 (Cell Signaling Technology (CST), #9411). EGFR was detected using rabbit anti-EGFR (Santa Cruz Biotechnology (SCBT), #sc-31157) and monoclonal rabbit anti-EGFR (CST, #4267) and immunoprecipitated using rabbit anti-EGFR sepharose beads (CST, #5735). p130CAS was detected using mouse anti-p130CAS (Becton-Dickinson, #610274) and immunoprecipitated using agarose beads conjugated with mouse anti-p130CAS (SCBT, #sc-20029). GAB1 was detected using mouse anti-GAB1 (Millipore, #06–579) and immunoprecipitated using the same antibody bound to Protein A sepharose CL-4B beads (GE Healthcare, #17–0780). SHCA was detected using rabbit anti-SHC (SCBT, #288) and immunoprecipitated using mouse anti-SHC (Thermo Scientific, #47F4) bound to Protein G agarose beads (Life Technologies, #20398). Phospho-SHCA was detected using rabbit anti-phospho-SHC pY317 (CST, #2431). EGFR phosphosite-specific Westerns were performed using the following antibodies: pY845 (SCBT, #sc-575442), pY974 (CST, #2641S), pY992 (CST, #2235P), pY1045 (CST, #2237P), pY1068 (CST, #3777P), pY1086 (CST, #2220S), and pY1173 (CST, #4407S).

Antibodies against pY1148 (CST, #4404S and Thermo Scientific, #44-792G) and pY1101 (Abcam, #ab76195) were tested but returned low signal to noise. GRB2 SH2-tdEOS and GST-GRB2-SH2 were detecting using mouse anti-GRB2 SH2 (R&D Systems, #669604). pERK1 and pERK2 were detected using rabbit anti p44/42 pT202/pY204 (CST, #9101S). Pervanadate (PV) was prepared fresh for each experiment by mixing 1 vol 100 mM $NaVO_4$ with 0.32 vol 30% (w/w) hydrogen peroxide at room temperature (RT) for 30–90 min.

## Sample preparation

All experiments were performed using the human squamous-cell carcinoma line A431. For FW and MS experiments, cells were grown to ~70% confluence in standard growth media, starved overnight and stimulated with 25 ng/mL EGF for the appropriate duration. Media was then removed and cells were snap frozen via submersion in liquid $N_2$ within 3 to 5 s. For imaging experiments, cells were plated onto acid-washed glass bottom dishes (<50% confluent) (MatTek) and allowed to grow overnight. Cells were transfected with 100–200 ng of DNA using 1–3 µl of Lipofectamine 2000 (Life Technologies) in antibiotic-free Opti-MEM (Life Technologies). After 4 hr, transfection medium was aspirated, replaced with complete medium and cells were allowed to grow overnight. Prior to imaging, DMEM culture medium was removed, cells were washed with PBS and kept in phenol red minus medium (BrainBits). For pervanadate experiments, pervanadate was prepared fresh, diluted in culture medium and added to cell culture dishes at a final concentration of 100–200 µM.

## Far-Western and Western blotting

Far-Western blotting was performed as previously described (*Machida et al., 2007*). Briefly, snap frozen cells were thawed on ice at 4°C, lysed and scraped in Kinase Lysis Buffer (150 mM NaCl, 25 mM Tris-HCl pH 7.4, 5 mM ethylene diamine tetraacetic acid (EDTA), 1 mM phenyl methyl sulfonyl fluoride (PMSF), 1% Triton X-100, 10% glycerol, 0.1% sodium pyrophosphate, 10 mM β-glycerophosphate, 10 mM NaF, 5 µg/ml of Aprotinin (Sigma A6279), 50 µM pervanadate), and cleared by centrifugation. 20 µg protein per lane was run on Lithium Dodecyl Sulfate (LDS) PAGE using NuPAGE NOVEX 4–12% gradient gels (Life Technologies, #WG1403A) and transferred overnight onto nitrocellulose membranes. 30 duplicate membranes were created, with all membranes containing positive and negative pY controls. Blots were frozen at -20°C, thawed, blocked with 5% non-fat dry milk in Tris pH 8.0 buffered saline (TBST) for 1 hr and blotted for 2 hr with 1–5 ng/ml recombinant GST-SH2 labeled with GSH-HRP and anti-GST-HRP diluted in 5% milk-TBST. Blots were then rinsed with TBST, washed for 20 min with two buffer changes and imaged using ECL (PerkinElmer, #NEL104001EA) on the Kodak Image Station 4000 MM for 1 hr. Reprobing was performed using the procedure outlined above, after stripping by washing with 100 mM glycine, pH 2.0 twice for 15 min followed by thorough washing with TBST. Blotting was performed at least twice for all quantified SH2 probes (technical replicates). Quantified far-Westerns met the following conditions: 1) pY-dependent binding (pervanadate-treated positive control sample has greater signal than PTP-treated negative control); 2) reproducibility for replicate membranes; 3) EGF-dependent changes in band intensity; 4) high signal-to-noise (minimal non-specific bands) and 5) high blot quality (minimal background and or signal distortion). 67 probes were initially screened for this study. Of these, the SH2 domains of BLK, BLNK, BRDG1, BRK, CBLB, CIS1, CSK, CTEN, EMT/ITK, FER, FES, FGR, FRK, GRB10, HCK, JAK3, LNK, LYN, P55G(NC), PLCG1(SH2+3), SH2-B, SH3BP2, SHB, SHIP1, SOCS2, SOCS3, SOCS4, SRC, STAT1, STAT3, STAT5A, SYK(NC), and ZAP70(NC) and PTB domains of CTEN, DOK1, FRS2, IRS1, SHCD, TENC1, and SCK, were excluded due to data quality. It should be noted that many of these probes have shown good activity in other systems (*Machida et al., 2007*; *2010*).

For Western blotting, nitrocellulose membranes were prepared in a manner similar to that outlined above for far-Western. Western blotting was performed using standard procedures (blocking in 5% milk, washes in TBST, overnight primary antibody incubation at 4o°C, 1 hr secondary antibody incubation at RT). Both HRP/ECL and IRDye (680 and 800 nm) labeled secondary antibodies were used. Blots were imaged using the Kodak Image Station 4000 MM or Licor Odyssey imager as appropriate.

## Protein identification by immunodepletion

SH2 binding proteins were tentatively identified via their SH2 binding affinities and molecular weights. Band identities were confirmed by repeated immunoprecipitation (3 or 6 times) of EGF-treated lysates using antibody-conjugated beads as listed above, followed by comparison of pre- and post-depletion lysates using the listed antibodies for immunoblot.

## SH2 binding band quantification and hierarchical clustering

Briefly, major protein bands of each blot were auto-detected and quantified using the Carestream MI software (Carestream) after subtracting background. Apparent non-specific bands with low reproducibility in multiple experiments were manually excluded. The molecular weight of each band was estimated using protein ladder lanes on both sides of the blot. Hierarchical clustering was performed using average linkage and un-centered correlation using Cluster 3.0 software. Heat maps were created using Java Treeview software. For kinetics clustering (*Figure 2B*), all bands from each probe were normalized to the band with the maximum signal in each replicate. Then data from each phosphoprotein was averaged to create a representative time course specific for each probe-phosphoprotein combination. Probes were then clustered based on the kinetics of probe binding site creation and their relative binding strength to each band. This is essentially a way to directly compare the overall far-western binding pattern for each probe. For relative specificity clustering (*Figure 2—figure supplement 3A*), raw FW quantification data for each band and time point (i.e. lane) were averaged. These averaged values were then expressed as a fraction of the total signal per lane. Lane specific fractional values from pY blots were then subtracted from the lane specific FW fractional values on a band-by-band basis.

## Determination of GRB2 SH2 and pY-EGFR concentration

The approximate cellular GRB2 SH2-tdEOS concentration was determined by comparing GRB2 SH2-tdEOS expression levels in transfected A431 cells with a GST-GRB2-SH2 standard of known concentration via immunoblot. Briefly, A431 cells (30 mm plate) were transfected with 1 µg GRB2 SH2-tdEOS using Lipofectamine 2000 (Life Technologies), incubated for 18 hr, lysed in KLB, run by LDS-PAGE along with a serial dilution of the GRB2 standard, transferred to nitrocellulose and immunoblotted with anti-GRB2 SH2. GRB2 SH2-tdEOS transfection efficiency and expression level distribution were determined by quantifying epifluorescence images from two parallel dishes acquired with a 1.5 min exposure using a CCD camera, using ImageJ. Average cell volume was calculated from three DIC images of non-adhered A431 cells. Cellular pY-EGFR concentrations were calculated by comparing pY immunoblots of EGF-stimulated cells with a tyrosine-phosphorylated protein standard created in our lab (derived from bacteria expressing a GST-ABL fusion protein) and quantified using a malachite green free phosphate quantification assay (*Thompson et al., 2015*). A431 cells were plated and EGF-stimulated using the protocol performed for FW above. Lysates were run on LDS-PAGE along with a serial dilution of the pY standard and immunoblotted using anti-pY. Signal intensity of the 195 kDa EGFR band and the total pY standard signal were quantified using Image Studio ver. 4.0 (Licor). The number of phosphotyrosines in the standard lanes were calculated and used to create a standard curve, which was then used to calculate the concentration of EGFR pY residues after correcting for the number of cells run per lane and the cell volume.

## Mass spectrometry sample preparation

Samples were lysed with 8 M urea + 1 mM sodium orthovanadate and protein yield was quantified by BCA assay (Pierce). Samples were reduced with 10 µl 10 mM DTT in 100 mM ammonium acetate pH 8.9 (1 hr at 56°C). Samples were alkylated with 75 µl of 55 mM iodoacetamide in 100 mM ammonium acetate pH 8.9 (1 hr at RT). 1 ml of 100 mM ammonium acetate and 10 µg of sequencing grade trypsin (Promega, #V5111) were added and digestion proceeded for 16 hr at RT. Samples were acidified with 125 µl of trifluoroacetic acid (TFA) and desalted with C18 spin columns (ProteaBio, #SP-150). Samples were lyophilized and subsequently labeled with iTRAQ 8plex (AbSciex) per manufacturer's directions. iTRAQ Channels were designated as follows: 0 s-113, 10 s-114, 30 s-115, 1 m-116, 1.5 m-117, 3 m-118, 10 m-119, 30 m-121.

## Immunoprecipitation

70 µl protein-G agarose beads (Calbiochem, #IP08) were rinsed in 400 µl IP Buffer (100 mM Tris, 0.3% NP-40, pH 7.4) and charged for 8 hr with three pY-specific antibodies: (12 µg 4G10 (Millipore), 12 µg PT66 (Sigma), and 12 µg PY100 [CST]) in 200 µl IP Buffer. Beads were rinsed with 400 µl of IP Buffer. Labeled samples were resuspended in150 µl iTRAQ IP Buffer (100 mM Tris, 1% NP-40, pH 7.4) + 300 µl milliQ water and pH was adjusted to 7.4 (with 0.5 M Tris-HCl pH 8.5). Sample was added to charged beads for overnight incubation. Supernatant was removed and beads were rinsed 3 times with 400 µl Rinse Buffer (100 mM Tris-HCl, pH 7.4). Peptides were eluted in 70 µl of Elution Buffer (100 mM glycine, pH 2) for 30 min at RT.

## Immobilized metal affinity chromatography (IMAC) purification

A fused silica capillary (FSC) column (200 µm ID x 10 cm length) was packed with POROS 20MC beads (Applied Biosystems, #1-5429-06). IMAC column was prepared by rinsing, at approximately 10 µl/min, with solutions in the following order: 100 mM EDTA pH 8.9 (10 min), $H_2O$ (10 min), 100 mM $FeCl_3$ (20 min), 0.1% acetic acid (10 min). IP elution was loaded at a flow rate of 2 µl/min. The column was rinsed with 25% acetonitrile, 1% acetic acid, and 100 mM NaCl (10 min) and 0.1% acetic acid (10 min), both at 10 µl/min. Peptides were eluted with 50 µl 250 mM $NaH_2PO_4$ at 2 µl/min and collected in an autosampler vial. Eluent was acidified with 2 µl of 10% TFA prior to loading.

## Liquid chromatography mass spectrometry

Acidified IMAC eluent was loaded onto an Acclaim PepMap 100 precolumn (Thermo Scientific, #164705) using an EASY-nLC 1000 (Thermo Scientific). Peptides were analyzed on a 1 hr gradient from 100% A (0.1% formic acid) to 100% B (0.1% formic acid, 80% acetonitrile) spraying through a 50 cm analytical column (New Objective, #PF360-50-10-N-5) packed with 3 µm beads (YMC America, #AQ12S03).

## MS data analysis

Thermo .RAW files were searched with MASCOT v2.4 using Proteome Discoverer (v1.4). Peptides that appeared in at least two biological replicates were included if their MASCOT scores exceeded 15 and they were designated as medium or high confidence by Proteome Discoverer. Normalized iTRAQ values for each biological replicate were averaged to produce the final dataset (*Supplementary file 2*, *MS Data*). Error is represented as the standard deviation (observed in all three biological replicates) or the average deviation (observed in two biological replicates).

## Gene ontology, sequence motif and EGF dependence analysis

Gene ontology (GO) analyses were performed using STRING 9.1. Biological Processes GO terms were queried and significant terms were identified as those with p-values ≤0.05 after Bonferroni correction for multiple comparisons. General gene ontologies used in the Venn diagram were obtained by GO semantic clustering using REVIGO (*Supek et al., 2011*). Sequence motif analysis was performed using the PhosphoSitePlus sequence logo generator using the frequency change algorithm and pY background (*Hornbeck et al., 2012*; *Vacic et al., 2006*). Previous documentation of EGF dependence was performed by comparing our data set with the PhosphositeSitePlus EGF-associated phosphorylation data set as of May 2015.

## Cell lines

A431 cells were originally obtained from Andrius Kazlauskas (Schepens Eye Research Institute, Boston MA). Cos1 cells were originally obtained from John Blenis (Harvard Medical School, Boston MA). H226 cells were obtained from the American Type Culture Collection (ATCC) as part of the NCI-60 panel of human cancer cell lines. A431 cells were periodically screened for mycoplasma contamination (STR authentication and current mycoplasma screening tests are ongoing and results will be provided on request).

## Apical and basal membrane pY quantification

A431 cells were grown to ~30% confluence in glass bottom 30 mm dishes and starved overnight in 0.1% FBS, 1% pen/strep DMEM. Cells were stimulated with 25 ng/mL EGF, quickly washed once in

4°C PBS and fixed with 4% paraformaldehyde on ice for 20 min (~15 s media-to-fix time). Cells were then washed with PBS three times for 5 min each, permeabilized in 1.5% Triton-X PBS for 10 min and washed with PBS three times for 10 min each. Dishes were then blocked in 1.5% BSA PBS for 60 min at 4°C, incubated with anti-pY (1:500 in blocking buffer) for 2 hr while rocking at 4°C, washed three times for 5 min in PBS, incubated with anti-mouse IgG conjugated with Alexa Fluor 594 at RT for 2 hr, washed three times for 5 min with PBS, covered with Fluoromount-G and a glass coverslip. Confocal microscopy was used to capture z-stacks through cells with significant apical-basal separation. Time points (min) and number of cells quantified are as follows: 0 (n=7), 0.08 (n=20), 0.17 (n=20), 0.5 (n=29), 1 (n=8), 2 (n=21), 4 (n=20), 6 (n=20), 10 (n=20) and 15 (n=20). For each cell, multiple line scans through the z-stack were quantified and averaged in both the x and y planes and used to plot the change in apical and basal phosphorylation in response to EGF.

## Cell proliferation assay

A431, Cos1, and H226 cells were seeded in 96-well plates at approximately 2,500 cells per well, cultured for 24 hr in the presence of 10% FBS, and then starved in 0.1% FBS medium overnight. Cells were then incubated with the 0.1% FBS medium with or without EGF at 0.1, 1, 25 ng/ml, or 10% FBS medium for three days. Cell proliferation was determined at 0 h, 24 hr, and 72 hr time points using CyQUANT direct assay according to manufacturer's instructions (Life Technologies). Briefly, equal volume of 2X detection reagent was added to the wells and incubated at 37°C for 1.5 hr. Fluorescence was measured at 508 nm excitation and 527 nm emission on an Infinite M1000 PRO plate reader (Tecan). Average fluorescence values of 16 replicated wells were normalized to the mean value for a set of wells at 0 hr and cell growth was determined as the fold change (Fluorescence_72 hr / Fluorescence_0 hr). Experiments were repeated at least three times and statistical differences were determined using multiple comparisons test of one-way ANOVA (Prism 6).

## TIRF and sptPALM microscopy

All live cell imaging were carried out on a total internal reflection (TIR) microscope custom built based on an Olympus IX81 microscope frame. All cells were kept at 37°C during the imaging experiments. For imaging all expressed SH2 molecules, a 488 nm Argon ion laser (Melles Griot) was used to detect tdEOS with a green filter set. The green channel data were used to quantify SH2 recruitment kinetics as well as the clustering kinetics as a function of EGF stimulation time. For sptPALM imaging, tdEOS was photoactivated using a 405 nm diode laser (Cube laser system, Coherent), excited with a 532 nm DPSS laser (Crystal Laser Inc), and detected with a red filter set. Fluorescence images were captured using a thermoelectric cooled EM CCD camera (PhotonMax). The sptPALM data were used to extract apparent binding and dissociation rates and diffusion constants of membrane-bound SH2 molecules.

## Single particle tracking

Individual diffraction-limited fluorescence dots displaying single step fluorescence photobleaching were localized using the standard method after application of a Gaussian kernel filter. Tracking over subsequent images was performed based on the nearest neighbor method. The scan area used during position tracking of individual molecules from frame to frame was chosen based on the estimated diffusion rate of the molecule. To avoid mistracking due to high particle density, the photoactivation was controlled to ensure a particle density of 0.1–0.05/$\mu m^2$ for all experiments. Mean square displacement (MSD) was calculated as a function of time lag using all trajectories over 4 frames long. Individual MSD-$\Delta t$ curves were fitted to the linear function to extract the diffusion constants (*Kusumi et al., 1993*). Unless noted in figure legend, D values were determined from 3000–20,000 trajectories in a single cell. Very little variation was seen in calculated D values for a particular SH2 domain in different cells.

## SH2 apparent on- and off-rate determination

The binding rate of GRB2 SH2 was determined based on sptPALM image sequences obtained at 10 Hz over a 0.8-min time duration. Newly emerged molecules in each video image sequence were counted after single particle tracking analysis. The apparent on-rate ($\gamma_{on}$) was calculated by dividing the number of newly emerged molecules by the total acquisition time (0.8 min). The apparent

membrane dissociation rate constant ($\lambda_{off}$) was determined using the trajectory length distribution from the same image sequences, as was described previously (*Oh et al., 2012*).

## Determination of clustered and non-clustered apparent on-rates

GRB2 SH2 clusters were identified by thresholding fluorescence images. The threshold intensity was determined computationally with the k-mean method. Thresholded binary images were used to characterize the number of clusters, cluster size and intensity (ImageJ). Thresholded binary images were also used as a mask to separate single-molecule trajectories into two groups – the clustered group and the non-clustered group – based on the initial localization of the trajectories. The number of molecules in each group was separately counted, giving apparent on-rates ($\gamma_{on}$) of clustered and non-clustered regions, respectively.

## Imaging EGF binding

A431 cells were seeded on glass substrate, serum starved-overnight, and imaged 12–18 hr after seeding using the same microscope as described in the TIRF imaging section, except the illumination incident angle was set to be just below the critical angle for total internal reflection, thus creating a thin illumination sheet traveling near the glass substrate surface. To assay EGF binding, TMR-labeled mouse EGF (ThermoFisher) was added directly to cell medium while time-lapse images were acquired at 5 s/frame with 561-nm laser excitation. Alternatively, imaging data were also acquired at 60 s/frame and compared with the 5 s/frame results in order to ensure that photobleaching is not a major factor distorting the intensity measurements. To quantify EGF binding, total integrated fluorescence signals from the whole cell were computed in ImageJ after background subtraction.

## Acknowledgements

This research was supported by the National Cancer Institute Grant U01CA154966 (to BJM and FW) and partly supported by a Quest for CURES (QFC) grant from the Leukemia and Lymphoma Society (to KM). We would like to thank Michael Blinov (UConn Health) for assistance with the SH2 membrane binding model and Ahmed Elmokadem for assistance with immunofluorescence imaging.

## Additional information

### Funding

| Funder | Grant reference number | Author |
| --- | --- | --- |
| National Institutes of Health | U01CA154966 | Forest M White<br>Bruce J Mayer |
| Leukemia and Lymphoma Society | QFC Grant 0818-14 | Kazuya Machida |

The funders had no role in study design, data collection and interpretation, or the decision to submit the work for publication.

### Author contributions

JAJ, DO, KM, JY, Conception and design, Acquisition of data, Analysis and interpretation of data, Drafting or revising the article; TGC, Acquisition of data, Analysis and interpretation of data, Drafting or revising the article; MO-I, Generated many of the constructs used, and helped draft and revise Supplementary file 1, Drafting or revising the article, Contributed unpublished essential data or reagents; LJ, Generated key preliminary data for FW blotting of EGF-stimulated cells, Drafting or revising the article, Contributed unpublished essential data or reagents; FMW, BJM, Conception and design, Analysis and interpretation of data, Drafting or revising the article

### Author ORCIDs

Bruce J Mayer, http://orcid.org/0000-0002-4580-3187

# Additional files

## Supplementary files

• Supplementary file 1. SH2 domain constructs. Cloning information and amino acid sequences for all cDNA constructs used in this analysis are listed.

• Supplementary file 2. Normalized FW, MS and in vivo imaging kinetics data. *Normalized Data-Interactive* tab provides a graph, which allows for the comparison of data from FW, MS, imaging and pY EGFR immunoblotting. To use, select the desired data from drop down menu. To remove data select 'Blank' located at the top of the menu. *Normalized Data* provides source data for the interactive chart (top) and all normalized data (below). Errors for FW and MS are shown as standard error of the mean and standard deviation, respectively. In vivo imaging and phosphosite specific western data are from single representative experiments. The interactive graph on the Normalized Data-Interactive tab and the data in Normalized Data tab are linked. The specific data sets used to plot curves of selected probes on the interactive graph can be found at the top of the Normalized Data tab. *FW Data* tab displays averaged binding quantifications for each SH2 probe. Data for each probe was normalized to the highest intensity band on each blot (Data used for *Figure 2B*). Error used is SEM. The number of (technical) replicate blots used is listed. *MS Data* tab provides sequence, Uniprot protein abbreviation and protein description for each peptide identified; indication of EGF dependence (two time points with Student's t-test p<0.05 and one time point with at least a two-fold increase compared to untreated samples); indication of sites not associated with EGF stimulation in PhosphoSitePlus database; and the number of biological replicates in which the peptides was detected. Phosphosite abundance data is normalized to sum of signal for all eight time points. Error is represented as standard or average deviation.

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
