## [Decision Letter]

Thank you for submitting your work entitled "Time-resolved multimodal analysis of SH2 domain binding in EGF-stimulated cells" for consideration by *eLife*. Your article has been favorably evaluated by Tony Hunter as Senior Editor and three reviewers, one of whom is a member of our Board of Reviewing Editors.

The reviewers have discussed the reviews with one another and the Reviewing Editor has drafted this decision to help you prepare a revised submission.

Summary:

The paper is a very interesting and comprehensive study of the interaction of different phosphotyrosine-binding SH2 domains to proteins that are phosphorylated in response to EGF in A431 cells. The combination of in vitro and in vivo approaches for following binding kinetics is a particular strength. The results show that different assays provide different and complementary results, and provide both a resource and a caution for other investigators. In this regard the paper makes an important and scholarly contribution. However, all reviewers were concerned that there was significant speculation and few answers as to why the different approaches yield different kinetics, and furthermore that some of the discrepancies may result from your use of cells that have extremely high receptor levels. It will be important to provide some data from more normal cells, that have a physiological response to EGF, or with a lower concentration of EGF in A431 cells, before the paper can be accepted.

Required revisions:

1) EGF binding and receptor occupancy should be measured over the time course and EGF concentration used in the paper. The results should be integrated into explanation of the kinetics of phosphorylation site kinetics and SH2 binding in vitro and in vivo. Appropriate discussion of the effects of receptor number and occupancy on the assays should be added.

2) Key experiments should be repeated in a cell line where EGF is mitogenic, or in A431 cells at a lower EGF concentration. The EGF concentration used in your experiments is reportedly growth inhibitory for A431 cells (Lifshitz et al. J Cell Physiol 115:235, 1983). It is important to test whether the discrepancies between different assays are also observed under mitogenic conditions.

3) Far Westerns should be probed with univalent SH2 domains, to test whether the different kinetics obtained with this method are due to avidity effects of the multivalent SH2 probes used.

4) The discussion of the effects of receptor clustering needs to incorporate the extensive literature on diffusion limited ligand binding, particularly with regard to the effects on on-rate of SH2 binding.

The full reviews now follow, and are included to help you understand the reasoning behind the required revisions.

*Reviewer #1:*

The authors have used iTRAQ MS, far Western blotting with purified SH2 domains, and single molecule imaging of SH2 domain-tdEOS fusion protein recruitment to the membrane, to monitor the kinetics of tyrosine phosphorylation events in A431 cells following EGF treatment. Each approach gives complementary results, with surprisingly different response kinetics. The authors discuss the limitations of each method and how together they help build a picture of the cellular response. The new findings include a transient dip in most MS detected sites 2-10 min after EGF. This may be due to recruitment of SH2-containing phosphatases. They also found discrepancies between the phosphorylation of individual sites detected by MS versus phospho antibodies. Perhaps most notably, SH2 binding sites, detected by Far Western, are apparently created more rapidly than SH2-tdEos fusions are recruited to the membrane. The latter may be limited by the time taken to form receptor clusters in the membrane.

The results are interesting in light of the many previous studies of EGF-stimulated tyrosine phosphorylation events. The paper may be a valuable resource for other investigators using this system. However, there are some major issues that weaken the results and lower the significance.

1) The authors used A431 cells, which have a hyper-abundance of EGFRs and are growth inhibited (not stimulated) at the EGF concentration used. This allowed them to follow more phosphorylation events but is of doubtful significance for cells that respond positively to EGF.

2) The responses detected by Far Western blotting were more rapid and had higher dynamic range than those detected by MS, and are faster than SH2-tdEos recruitment in vivo. This may in part be due to the use of clustered SH2 domains to probe the blots (GST-SH2, anti-GST-HRP and GSH-HRP), which will increase avidity and likely cause marked non-linearity in binding, amplifying the signal. Controls with monovalent SH2 domains should be done. The phosphoprotein targets are of course also denatured. It might be more informative to follow association of the endogenous full length proteins by conventional co-IP.

3) The membrane recruitment of fluorescent SH2 domains in vivo was analyzed both by conventional TIRF, providing on- and off-kinetics for net membrane association, and by single molecule PALM, allowing measurement of the number of molecules binding and releasing per unit time, cluster size and diffusion in the plane of the membrane (related to clustering). Based on this they conclude that SH2 domains only bind stably to clustered receptors (due to rebinding and hence low off rate), and that the slow formation of clusters slows down membrane association. This principle was already investigated in the authors' nice PNAS paper (Oh et al. 2012). However, the results don't exclude the possibility that the on-rate to EGFR clusters may also be low. Since EGFR clustering leaves more EGFR-free membrane area, a decrease in on-rate is not unreasonable. It may be possible to measure on-rate to a cluster by photo-converting a cluster and looking at new molecule binding.

*Reviewer #2:*

This is a very interesting and comprehensive study of the interaction of different phosphotyrosine-binding SH2 domains to proteins phosphorylated in response to EGF addition to A431 cells. The paper is technically excellent and is very well written. I had no trouble following the logic of both the experiments and the interpretations. The strengths of the paper include its use of multiple technical approaches to follow both protein phosphorylation and the dynamic interaction of different SH2 domains to their targets, its comprehensive nature and the excellent data sets included with this study (Table 1 makes the entire study worthwhile, in my opinion). The combination of in vitro and in vivo approaches for following binding kinetics is a particular strength.

Where this paper falls short is in placing their results into the context of what is already known and in understanding the consequences of using A431 cells, which have enormous numbers of EGFR.

Their basic conclusion is that clustering of the EGFR affects the ability of SH2-domain proteins to bind and dissociate from the membrane. If the receptors are diffusely distributed, binding is fast. If they are clustered, it is delayed. The reverse rate is also slower if the receptors are clustered. Both of these effects of high receptor density or clustering can be due to diffusion-limited binding, which has been well studied since the seminal work of Berg and Purcell over 35 years ago (Biophy J., 20:193-219, 1977). Although previous studies on the effect of high receptor density and clustering have focused on ligands, binding of SH2-domains should be similar. Indeed, those studies have described the same sort of effects that are described here (e.g., Potanin et al., [1994] Eur Biophys J 23, 197; Lagerholm and Thompson [1998] Biophys J 74, 1215; Gopalakrishnan et al., [2005] Biophys J 89, 3686). For example, although the "apparent binding rates" of SH2-domain proteins to clustered receptors are reduced relative to unclustered receptors, this is because clustered sites compete with each other for binding to diffusing ligands. Dissociation rates look slower because of rebinding effects, and so forth. Diffusion-limited binding of EGF has even been explored in A431 cells because of their extraordinary number of EGFR (see Wiley [1988] J Cell Biol 107, 801). I am not suggesting that these previous studies make the findings presented here superfluous, but they have direct relevance for interpreting their results, especially the effect of clustered density versus total receptor density and the effect of time-dependent receptor occupancy versus time-dependent SH2-domain protein binding. For example, because of diffusion-limited binding of EGF to A431 cells, total receptor occupancy will climb continuously over time rather than rapidly reaching equilibrium (see Figure 2 of Wiley [1988]). This looks very similar to the continuous increase in specific EGFR phosphorylation over time in Figure 3. The "oscillation" they report in the tyrosine phosphorylation levels after stimulation with EGF is thus likely due to a rapid desensitization of the initially activated receptors followed by a slow accumulation of additionally occupied receptors.

Because of diffusion-limited binding, there are three processes they must untangle: 1) the effect of a continuously increasing number of occupied EGFR over time, 2) the time-dependent change in the overall density of activated receptors at the cell surface and its effect on receptor clustering, and 3) the density of receptors in the clusters. All three of these processes will be dependent on the levels of occupied receptors, which unfortunately, were not examined in this paper. Without knowing this information, it is difficult to know whether the types of effects seen here (e.g. Figure 5) are also present in cells lacking EGFR amplification. It’s not necessarily a bad thing if these effects are only seen in EGFR-amplified cells, but this knowledge would certainly help the reader know how widely their results can be extrapolated to other systems. I would suggest that they quantify the time-dependent change in EGFR occupancy in their cells using fluorescently-labeled EGF (available commercially) and show that the range of receptor occupancy they are observing is within the range of other cells. Alternately, they could show that doses of EGF that result in lower levels of net occupancy produce similar results. At the very least, they need to incorporate the previous literature on diffusion-limited binding processes in their Discussion.

*Reviewer #3:*

Jadwin et al. present a comprehensive analysis of EGF-stimulated phosphorylation and associated SH2/PTB interaction datasets, and was a pleasure to read. The model cell system is the cancer-derived cell line A431, which is known to express high level wild type EGFR (>1-million per cell). The authors are experienced and credible in the field, and in this instance sought to address outstanding questions related to SH2/PTB-mediated signaling downstream of activated EGFR, but generally relevant to tyrosine kinase signaling. Far western analysis showed positively correlated kinetics of binding of a set of domains to proteins known to be tyrosine phosphorylated in response to EGFR activation. These data are complemented by MS-based analysis of individual phospho-peptides, and by measurement of membrane recruitment of ectopically expressed domains. Overall, the authors present interesting data, and Table 1 stands out as an engaging, interactive tool. A main conclusion drawn is that the integration of orthogonal datasets is an effective approach to reveal discrepancies that may relate to technical limitations or possibly biological mechanisms.

In some instances the presentation and discussion of data lack clarity, and there is considerable speculation based on technical phenomenology.

1) The coverage of EGFR phosphorylation sites, central to the study, was not as exhaustive as many published reports, and was not clearly acknowledged/presented or discussed. For example, they describe five different EGFR phosphopeptides, but three are phospho-isomers that contain pY1045. A peptide singularly phosphorylated at the primary GRB2 SH2 site pY1068 was not part of the data, which compromises the interpretation of results and discussions about GRB2.

2) Obviously FW and MS analyses, by technical design, measure different features, but the assumptions associated with these methods should be more thoroughly addressed.

3) Some conclusion statements reiterate consensus views, and hence would be appropriate for the Introduction. Examples include:

"…suggests that the concentration of SH2 domain binding sites can be as important as SH2 domain binding specificity…"

"…suggest that the stoichiometry of SH2 and PTB domain-containing effectors bound to EGFR is temporally regulated by differential phosphorylation/dephosphorylation of their specific binding sites…"

"…these data suggest that EGFR overexpression is associated with a significant expansion of its classical downstream signal transduction pathways"

4) What are the estimated concentrations of ectopic domains versus endogenous? For example, is competition between ectopic GRB2 SH2 with endogenous GRB2, which is reportedly highly expressed (approx. 500K/Hela cell), factored into the calculations and data interpretation?

5) The interpretation that the pervanadate effect is due to a lack of clustering is speculative. What is the (enhanced) stoichiometry/level of phosphorylation in response to pervanadate compared with EGF?

---

## [Author Response]

Required revisions:

*1) EGF binding and receptor occupancy should be measured over the time course and EGF concentration used in the paper. The results should be integrated into explanation of the kinetics of phosphorylation site kinetics and SH2 binding in vitro and in vivo. Appropriate discussion of the effects of receptor number and occupancy on the assays should be added.*

As suggested, we have performed binding experiments with labeled EGF and now present these data (Figure 4—figure supplement 2). EGF binding was quite rapid under conditions used to monitor intracellular responses, and parallels tyrosine phosphorylation of the EGFR as expected. We now explicitly discuss the contribution of EGF binding to the kinetics of receptor output in the eighth paragraph of the Discussion section.

*2) Key experiments should be repeated in a cell line where EGF is mitogenic, or in A431 cells at a lower EGF concentration. The EGF concentration used in your experiments is reportedly growth inhibitory for A431 cells (Lifshitz et al. J Cell Physiol 115:235, 1983). It is important to test whether the discrepancies between different assays are also observed under mitogenic conditions.*

We have repeated key experiments (in vivo SH2 binding, far-Western blotting) with the GRB2 SH2 domain probe to confirm that the apparent lag in reaching maximal binding in vivo seen at high EGF concentrations (25ng/ml) is also seen at lower EGF concentrations (1ng/ml), where total pTyr levels are considerably lower and where EGF is mitogenic in A431 cells. These results are now shown in Figure 4—figure supplement 6. See below for further details.

*3) Far Westerns should be probed with univalent SH2 domains, to test whether the different kinetics obtained with this method are due to avidity effects of the multivalent SH2 probes used.*

We addressed whether conditions of FW blotting may have resulted in non-linear response (that is, signal was saturated at high levels of tyrosine phosphoryated protein on the filter, thereby obscuring further increases in phosphorylation) by performing FW blotting experiments with serial dilutions of the same samples on the same blot (Figure 4—figure supplement 4). We found that the response is close to linear over a wide range of concentrations, and that at all concentrations of analyte tested, it was possible to detect higher levels of phosphorylation (i.e. the assay was not saturated). We feel these experiments definitively rule out any technical artifact of the far-Western blotting method that could lead to a failure to detect increases in SH2 binding sites after the first minute or two.

*4) The discussion of the effects of receptor clustering needs to incorporate the extensive literature on diffusion limited ligand binding, particularly with regard to the effects on on-rate of SH2 binding.*

We now have included a discussion of diffusion-limited ligand binding and its possible effects on the on-rate of SH2 binding (Discussion, paragraphs eight to ten). Please see our response to reviewer 2, below for a more detailed explanation.

*The full reviews now follow, and are included to help you understand the reasoning behind the required revisions.*

Reviewer #1:

*[…] The results are interesting in light of the many previous studies of EGF-stimulated tyrosine phosphorylation events. The paper may be a valuable resource for other investigators using this system. However, there are some major issues that weaken the results and lower the significance. 1) The authors used A431 cells, which have a hyper-abundance of EGFRs and are growth inhibited (not stimulated) at the EGF concentration used. This allowed them to follow more phosphorylation events but is of doubtful significance for cells that respond positively to EGF.*

The reviewer raises an excellent point, that the kinetics observed in A431 cells treated with high concentrations of EGF might not reflect those at more physiological levels of receptor and/or ligand. As noted above, we have now confirmed that our results are similar (including the discrepancy between timing of maximal phosphorylation and maximal SH2 binding in vivo) in A431 cells treated with much lower EGF concentrations (1ng/ml vs. 25ng/ml used previously). This concentration is mitogenic (Figure 4—figure supplement 5) in our A431 cells, and results in only ~20% of the phosphorylation of EGFR seen at 25ng/ml EGF (Figure 4—figure supplement 6). This issue is now explicitly discussed in the revised manuscript (Discussion, twelfth paragraph). We hope this addresses the reviewer’s concern.

Unfortunately, at 1ng/ml EGF, the in vivo membrane recruitment levels were close to background using our imaging system, but the GRB2 SH2 gave a relatively robust and reproducible signal. We also note that virtually all published MS-based phosphoproteomic studies on EGFR signaling have used very high concentrations of EGF (the same or higher than we used, 25 ng/ml), and a number have used A431 cells as well. Furthermore, we show that the major mitogenic output out EGFR signaling, activation of the MAPK ERK1/2, is stimulated with kinetics comparable to those seen other published studies/systems when A431 are stimulated by 25ng/ml EGF (Figure 2—figure supplement 2). For these reasons, we feel that our detailed phosphoproteomic studies are a valuable contribution to our understanding of RTK signaling dynamics, even if most data are obtained from cells treated with high concentrations of EGF.

*2) The responses detected by Far Western blotting were more rapid and had higher dynamic range than those detected by MS, and are faster than SH2-tdEos recruitment in vivo. This may in part be due to the use of clustered SH2 domains to probe the blots (GST-SH2, anti-GST-HRP and GSH-HRP), which will increase avidity and likely cause marked non-linearity in binding, amplifying the signal. Controls with monovalent SH2 domains should be done. The phosphoprotein targets are of course also denatured. It might be more informative to follow association of the endogenous full length proteins by conventional co-IP.*

As noted above, we have now directly demonstrated in new experiments that potential non-linearity of the FW assay cannot explain the lack of apparent increase in GRB2 SH2 binding to EGFR after 1-2 minutes (Figure 4—figure supplement 4). We should add that antibodies, being divalent, have the same potential issues with avidity and nonlinearity as the dimeric GST-SH2 domains used as probes in FW blots. We did not repeat FW experiments with monovalent SH2 domains because dimerization is required for stable SH2 binding to blots (see Nollau and Mayer, PNAS 2001; 98:13531-13536), as expected since the off-rates for monomeric SH2 domains are very rapid (>> 1/sec for GRB2 SH2 domain). But in any case, we hope that the concern about nonlinearity was satisfactorily addressed by these new experiments.

*3) The membrane recruitment of fluorescent SH2 domains in vivo was analyzed both by conventional TIRF, providing on- and off-kinetics for net membrane association, and by single molecule PALM, allowing measurement of the number of molecules binding and releasing per unit time, cluster size and diffusion in the plane of the membrane (related to clustering). Based on this they conclude that SH2 domains only bind stably to clustered receptors (due to rebinding and hence low off rate), and that the slow formation of clusters slows down membrane association. This principle was already investigated in the authors' nice PNAS paper (Oh et al. 2012). However, the results don't exclude the possibility that the on-rate to EGFR clusters may also be low. Since EGFR clustering leaves more EGFR-free membrane area, a decrease in on-rate is not unreasonable. It may be possible to measure on-rate to a cluster by photo-converting a cluster and looking at new molecule binding.*

This is a good point, and actually was one of our initial hypotheses to explain the discrepancy. However, our attempts to test this directly by FRAP (photobleaching a cluster and looking at recovery of fluorescence, to allow estimation of on-rate to clusters) were inconclusive at best, due to difficulties in fitting the experimental recovery curves. The experiment suggested by the reviewer (photo-activating a cluster) is even more challenging, as rapid diffusion of any photoactivated probe away from the cluster and mixing with the vast excess of probe in the cytosol would prevent reliable measurement of on-rates. We therefore turned to the experiments shown in Figure 5, which showed that the aggregated on-rate for clustered regions increased in concert with the number and size of clusters, whereas there was a dramatic decrease in the off-rate as clustering proceeded. These results are most consistent with the idea that it is decreased off-rate, and not decreased on-rate, that is most important in slowing the approach to maximal binding in vivo. See also our response to reviewer 2, below.

Reviewer #2:

*[…]* Where this paper falls short is in placing their results into the context of what is already known and in understanding the consequences of using A431 cells, which have enormous numbers of EGFR.

Please see response to general comment 2, and reviewer #1, point 1 above.

*Their basic conclusion is that clustering of the EGFR affects the ability of SH2-domain proteins to bind and dissociate from the membrane. If the receptors are diffusely distributed, binding is fast. If they are clustered, it is delayed. The reverse rate is also slower if the receptors are clustered. Both of these effects of high receptor density or clustering can be due to diffusion-limited binding, which has been well studied since the seminal work of Berg and Purcell over 35 years ago (Biophy J., 20:193-219, 1977). Although previous studies on the effect of high receptor density and clustering have focused on ligands, binding of SH2-domains should be similar. Indeed, those studies have described the same sort of effects that are described here (e.g., Potanin et al., [1994] Eur Biophys J 23, 197; Lagerholm and Thompson [1998] Biophys J 74, 1215; Gopalakrishnan et al., [2005] Biophys J 89, 3686). For example, although the "apparent binding rates" of SH2-domain proteins to clustered receptors are reduced relative to unclustered receptors, this is because clustered sites compete with each other for binding to diffusing ligands. Dissociation rates look slower because of rebinding effects, and so forth. Diffusion-limited binding of EGF has even been explored in A431 cells because of their extraordinary number of EGFR (see Wiley [1988] J Cell Biol 107, 801). I am not suggesting that these previous studies make the findings presented here superfluous, but they have direct relevance for interpreting their results, especially the effect of clustered density versus total receptor density and the effect of time-dependent receptor occupancy versus time-dependent SH2-domain protein binding. For example, because of diffusion-limited binding of EGF to A431 cells, total receptor occupancy will climb continuously over time rather than rapidly reaching equilibrium (see Figure 2 of Wiley (1988)). This looks very similar to the continuous increase in specific EGFR phosphorylation over time in Figure 3. The "oscillation" they report in the tyrosine phosphorylation levels after stimulation with EGF is thus likely due to a rapid desensitization of the initially activated receptors followed by a slow accumulation of additionally occupied receptors.*

*Because of diffusion-limited binding, there are three processes they must untangle: 1) the effect of a continuously increasing number of occupied EGFR over time, 2) the time-dependent change in the overall density of activated receptors at the cell surface and its effect on receptor clustering 3) the density of receptors in the clusters. All three of these processes will be dependent on the levels of occupied receptors, which unfortunately, were not examined in this paper. Without knowing this information, it is difficult to know whether the types of effects seen here (e.g. Figure 5) are also present in cells lacking EGFR amplification. It’s not necessarily a bad thing if these effects are only seen in EGFR-amplified cells, but this knowledge would certainly help the reader know how widely their results can be extrapolated to other systems. I would suggest that they quantify the time-dependent change in EGFR occupancy in their cells using fluorescently-labeled EGF (available commercially) and show that the range of receptor occupancy they are observing is within the range of other cells. Alternately, they could show that doses of EGF that result in lower levels of net occupancy produce similar results. At the very least, they need to incorporate the previous literature on diffusion-limited binding processes in their Discussion.*

We thank the reviewer for these thoughtful comments. The main concern is how the kinetics of ligand binding affect the intracellular processes described in our studies. In the revision, we tried to address this complicated question from several different angles.

First we characterized EGF binding kinetics in the same A431 cells and conditions we used for other experiments. We used a real-time florescence imaging assay. In short, we added fluorescently-labeled EGF to adherent cells and imaged the cells with an oblique illumination beam, which eliminates most of the background signal due to the soluble EGF, yet still allows excitation of the whole cell. This approach was originally demonstrated by the Sako group (see e.g. EMBO J 2006; 25:4215) for studying EGF-EGFR complexes. As shown in Figure 4—figure supplement 2, at 25ng/ml EGF the binding of EGF saturated relatively quickly at around 1 min, and in fact the signal decreased slightly after that. Therefore, we do not think that in our system the number of occupied EGFR continuously increased over time. We note that this is a somewhat different observation than Wiley (1988),which reported a continuous increase of EGF binding even after 10 min. One potential explanation for the discrepancy is that Wileymeasured binding to a confluent monolayer of cells, while our experiments were performed at much lower density. It is also possible that differences in the EGFR density or endocytosis rate in the specific A431 cells used in the different studies could contribute to the discrepancy.

The second question is whether the ligand binding kinetics are of any indicative value towards understanding intracellular signaling process. At 25ng/ml, the EGF binding kinetics are fairly consistent with the time-frame we see for achieving maximal phosphorylation of the receptor by anti-pY western or SH2 FW blotting. We do note that our experimental data (FW, MS, imaging) all deal with the dynamics of intracellular events in EGFR signaling (that is, phosphorylation of the receptor and its substrates, receptor clustering, and the binding of effector SH2 domains to the phosphorylated receptor and substrates). Rates of EGF binding, while of obvious importance in triggering EGFR phosphorylation (and ultimately clustering), are therefore not directly informative with respect to discrepancies between kinetics of phosphorylation and SH2 binding.

The last question is whether the delayed binding of SH2 effectors should be understood mechanistically in terms of diffusion-limited reactions. First of all, there is no question that diffusion-limited reaction kinetics plays an important role in governing receptor-effector interactions. We have analyzed such dynamics in detail in our previous publication (PNAS 109:14024). That said, it may be presumptuous to claim that the slow binding of SH2 domain is solely due to the diffusion-limited kinetics. The diffusion-limited on-rate, in its simplest form, e.g., according to Berg and Purcell (Biophy J.1977; 20:193-219),roughly scales with 4πDs, where *s* is a reaction cross-section. Based on what we know about the SH2 concentration, the diffusion constant and best estimate of receptor density in clusters, the theoretical prediction would suggest a delay of at best tens of seconds, not multiple minutes after the intracellular pY concentration reached its peak. The simple analysis of course is not conclusive because it does not take into account spatial heterogeneity and, more importantly, competition between different SH2 binding targets. However, the point is that there is no simple intuitive argument that can be convincingly made to either validate or rule out diffusion-limited kinetics as a mechanistic explanation. A more thorough mathematical model is beyond the scope of this paper. Nevertheless, we did add an extensive discussion on this topic (Discussion section) to communicate our interpretations more clearly and in the context of existing theory and previous work, as was suggested by the reviewer.

Reviewer #3:

*[…] In some instances the presentation and discussion of data lack clarity, and there is considerable speculation based on technical phenomenology.*

We have revised the text in an attempt to increase clarity and minimize unfounded speculation.

*1) The coverage of EGFR phosphorylation sites, central to the study, was not as exhaustive as many published reports, and was not clearly acknowledged/presented or discussed. For example, they describe five different EGFR phosphopeptides, but three are phospho-isomers that contain pY1045. A peptide singularly phosphorylated at the primary GRB2 SH2 site Y1068 was not part of the data, which compromises the interpretation of results and discussions about GRB2.*

We agree that the coverage of the EGFR phosphorylation sites was not as comprehensive as several other published reports, including some of the reports previously published from the White lab. However, as the reviewer may appreciate, phosphorylation site identification and quantification may be limited by the dynamic range and stoichiometry of each phosphorylation site as well as the nature of the mass spectrometry experiment. In this case, the mass spectrometry-based phosphorylation analysis was conducted in data-dependent, discovery-mode to provide the greatest network coverage in a relatively unbiased fashion (as opposed to more directed approaches that focus on specific, targeted ions). In general, this analysis typically identifies the most abundant phosphorylation sites, as the selection of ions is based on intensity in each full scan mass spectrum combined with the exclusion list information. As such, the lack of detection of some of the EGFR phosphorylation sites may be due to their low level of phosphorylation relative to other sites in the network or even on the receptor. The identification and quantification of three isoforms of the peptide contained EGFR Y1045/1068 indicates that this peptide could be detected and quantified. Since the mono-phosphorylated (Y1068) peptide was not detected, it is likely that this phosphorylation site more often occurred in concert with Y1045 phosphorylation, and that the Y1068 mono-phosphorylated peptide was therefore much lower in intensity compared to the other three isoforms. Given the likely relatively low level of this mono-phosphorylated peptide, or the other phosphorylation sites on EGFR not reported in our data, we do not feel that this missing information adversely affects the interpretation of the data.

*2) Obviously FW and MS analyses, by technical design, measure different features, but the assumptions associated with these methods should be more thoroughly addressed.*

We had tried to discuss the assumptions in the previous version, but have now modified the text to address this issue more thoroughly (see for example subsection “Quantitative phospho-specific mass spectrometry”).

*3) Some conclusion statements reiterate consensus views, and hence would be appropriate for the Introduction. Examples include:*

*"…suggests that the concentration of SH2 domain binding sites can be as important as SH2 domain binding specificity…" "…suggest that the stoichiometry of SH2 and PTB domain-containing effectors bound to EGFR is temporally regulated by differential phosphorylation/dephosphorylation of their specific binding sites…" "…these data suggest that EGFR overexpression is associated with a significant expansion of its classical downstream signal transduction pathways"*

We agree that each of these statements represents consensus views, and have modified the text to more clearly convey this fact.

*4) What are the estimated concentrations of ectopic domains versus endogenous? For example, is competition between ectopic GRB2 SH2 with endogenous GRB2, which is reportedly highly expressed (approx. 500K/Hela cell), factored into the calculations and data interpretation?*

We quantified the levels of endogenous and exogenous GRB2 in these cells in Figure 4—figure supplement 3). Expression levels of exogenous SH2 varies from cell to cell, but as noted in the text, the average concentration is 6.5μM. We also now state in the figure legend the endogenous level of GRB2 in these cells, 1.5μM.

*5) The interpretation that the pervanadate effect is due to a lack of clustering is speculative. What is the (enhanced) stoichiometry/level of phosphorylation in response to pervanadate compared with EGF?*

As shown in Figure 6, by both anti-pTyr and GRB2 SH2 FW, the extent of phosphorylation of EGFR in pervanadate-treated cells greatly exceeds that of EGF-treated cells at later time points. Note that since these lysates were run on the same blot, the extent of phosphorylation in different samples can be directly compared. It is not possible to evaluate stoichiometry (distinct from level) of phosphorylation in these experiments.